# Circular RNA *circTmem241* drives group III innate lymphoid cell differentiation via initiation of *Elk3* transcription

Nian Liu[1,2,5], Jiacheng He[1,2,5], Dongdong Fan[3,5], Yang Gu[1,2], Jianyi Wang[1,2], Huimu Li[1,2], Xiaoxiao Zhu[3], Ying Du[1], Yong Tian [2,3] ✉, Benyu Liu [4] ✉ & Zusen Fan [1,2] ✉

Innate lymphoid cells (ILCs) exert important roles in host defense, tissue repair and inflammatory diseases. However, how ILC lineage specification is regulated remains largely elusive. Here we identify that circular RNA *circTmem241* is highly expressed in group III innate lymphoid cells (ILC3s) and their progenitor cells. *CircTmem241* deficiency impairs ILC3 commitment and attenuates antibacterial immunity. Mechanistically, *circTmem241* interacts with Nono protein to recruit histone methyltransferase Ash1l onto *Elk3* promoter in ILC progenitor cells (ILCPs). Ash1l-mediated histone modifications on *Elk3* promoter enhance chromatin accessibility to initiate *Elk3* transcription. Of note, *circTmem241*[−/−], *Nono*[−/−] and *Ash1l*[−/−] ILCPs display impaired ILC3 differentiation, while Elk3 overexpression rescues ILC3 commitment ability. Finally, *circTmem241*[−/−]*Elk3*[−/−] mice show lower numbers of ILC3s and are more susceptible to bacterial infection. We reveal that the *circTmem241*-Nono-Ash1l-Elk3 axis is required for the ILCP differentiation into ILC3P and ILC3 maturation, which is important to manipulate this axis for ILC development on treatment of infectious diseases.

Innate lymphoid cells (ILCs) are heterogeneous populations of lymphocytes that react rapidly to environmental cues and exert critical roles in regulating mucosal immunity, lymphoid organogenesis, and tissue homeostasis[1]. ILC subsets mirror T helper (Th) cells based on transcription factors (TFs) and secreted cytokines. ILC1s express T-bet and secret IFN-γ to function in the immune response against intracellular pathogens[2]. ILC2s express Gata3 and secret Th2-like cytokines such as IL-5 and IL-13, which play an important role in allergic response and innate immunity to helminth[3]. ILC3s express RORγt and mainly produce IL-22 and/or IL-17 after activation[4]. All ILC subsets are developed from common lymphoid progenitors (CLPs), which give rise to common innate lymphoid progenitors (CILPs), then diverge into common helper innate lymphoid progenitors (CHILPs) and NK cell

precursors (NKPs). CHILPs can then differentiate into lymphoid tissue inducer progenitors (LTiPs) and innate lymphoid cell precursors (ILCPs), which finally produce all groups of ILCs[5,6]. The commitment of ILCs is finely modulated by fate-decision TFs. Expression of inhibitor of DNA binding 2 (Id2) is required for establishing ILC cell fate from CLPs by repressing E-box protein activity[7,8]. Id2[+] CHILPs give rise to more restricted ILCPs via expression of transcription factor PLZF (encoded by *Zbtb16* gene) and possess the ability to produce all ILC subsets[7,8]. Recent studies have identified specific precursors of each ILC subsets[9,10]. ILC1 precursors (ILC1Ps, Lin[−]CD127[+]Eomes[−]CD49a[+]NK1.1[+]NKp46[+]) require T-bet, Nfil3 and Gata3 for ILC1 differentiation[11]. Transcription factors such as Bcl11b and Gata3 are involved in ILC2 commitment from ILC2 precursors (ILC2Ps, Lin[−]CD127[+]CD45[+]Flt3[−]CD117[−]Sca-1[+]CD25[+]Gata3[+]) to

[1]Key Laboratory of Infection and Immunity of CAS, CAS Center for Excellence in Biomacromolecules, Institute of Biophysics, Chinese Academy of Sciences, Beijing 100101, China. [2]University of Chinese Academy of Sciences, Beijing 100049, China. [3]Key Laboratory of RNA Biology, Institute of Biophysics, Chinese Academy of Sciences, Beijing 100101, China. [4]Research Center of Basic Medicine, Academy of Medical Sciences, Zhengzhou University, Zhengzhou, Henan, China. [5]These authors contributed equally: Nian Liu, Jiacheng He, Dongdong Fan. ✉e-mail: ytian@ibp.ac.cn; benyuliu@zzu.edu.cn; fanz@moon.ibp.ac.cn

mature ILC2s[9,12,13]. Whereas ILC3 precursors (ILC3Ps, Lin$^-$CD25$^-$CD127$^+$α$_4$β$_7^{int}$ Gata3$^{lo}$Bcl11b$^+$Id2$^+$RORγt$^+$) drive ILC3-restricted lineage specification[14]. ILC3 commitment and function are finely regulated by various aspects. Extrinsic clues and TFs like Gata3 and Rorc are directly involved in ILC3 fate decision[15–17]. We previously showed that *lncKdm2b* activates *Zfp292* expression to promote the maintenance and effector functions of ILC3s[18]. We also found that IL-7Rα glutamylation is required for ILC3 specification[19].

Circular RNAs (circRNAs) are generated from precursor mRNA through back-splicing events, which are characterized by a covalent bond linking the 3′- and 5′-ends[20]. CircRNAs comprise exons, introns, or both. Compared to linear RNAs, circRNAs have a longer half-life and resist exonuclease degradation[21,22]. CircRNAs exist in various types of tissues and cells and exert critical roles in many biological processes, including development, stemness maintenance, and tumorigenesis. For example, circRNA *Cdr1as* regulates brain function via sponging miR-7 and miR-671[23]. Fusion circRNAs derived from transcribed exons of some genes promote tumorigenesis and enhance resistance against therapy[24]. A recent report showed that endogenous circRNAs may form imperfect RNA duplexes to suppress activation of PKR, resulting in inhibition of innate immunity[25]. We previously demonstrated that circRNA *cia-cGAS* is highly expressed in hematopoietic stem cells (HSCs) and maintains HSC homeostasis via blocking nuclear cGAS enzymatic activity[26]. We also showed that circRNA *circKcnt2* facilitates colitis resolution by inhibiting ILC3 activation[27]. However, how circRNAs regulate ILC differentiation still remains elusive.

ETS domain-containing protein Elk3 (Elk3) belongs to the ETS-domain transcription factor family and the ternary complex factor (TCF) subfamily. Elk3 is well known to regulate the early response to growth factor stimulation in quiescent cells[28]. Elk3 is a strong transcriptional repressor and is involved in several biological processes such as angiogenesis and tumorigenesis[29]. A previous study showed that phosphorylation of Elk3 switches it to a transcriptional activator, leading to induction of VEGF expression and angiogenesis[28]. Elk3 can regulate chromatin landscape and initiate the progression of squamous cell carcinomas[30]. Importantly, Elk3 also plays an essential role in the development of neural cells[31]. Moreover, several studies revealed that Elk3 is a potential regulator of immune response[32,33]. Elk3 inhibits HO-1 and NOS2 expression under LPS or cytokine stimulation and regulates HO-1-mediated inflammatory response in macrophages[32,34]. In addition, Elk3 displays different expression patterns in CD4$^+$ and CD8$^+$ T cells that regulate immune response mediating tumorigenesis of colon cancer[35]. However, the role of Elk3 in ILC biology is still unclear. Here we identified an undefined circular RNA *circTmem241* (derived from *Tmem241* gene transcript, mmu_circ_0007131 in circBase) that is highly expressed in ILC3s and their progenitor cells CHILPs and ILCPs. *CircTmem241* regulates ILC3 specification at the ILCP stage, and the *circTmem241*-Nono-Ash1l-Elk3 axis is required for ILC3 differentiation.

## Results

### *CircTmem241* knockout reduces ILC3 numbers

We previously performed circRNA microarray analysis of ILC3s from *Rag1$^{-/-}$* mice and identified differentially expressed circRNAs in ILC3s related to innate colitis[27]. To explore the role of circRNAs in ILC3 commitment, we selected the top 10 highly expressed circRNAs in *Rag1$^{-/-}$* ILC3s that were conserved between mice and humans, whose circular characteristics were validated by PCR (Supplementary Fig. 1a, b), and Sanger sequencing (Supplementary Fig. 1c). These ten circRNAs were resistant to RNase R digestion (Supplementary Fig. 1d). We next knocked them down in CHILPs using lentivirus (Supplementary Fig. 1e) and conducted in vitro differentiation assay. Knockdown of circRNAs did not affect their expression levels of their parental genes (Supplementary Fig. 1f). Of these ten circRNAs, *circTmem241* (circBase ID:

mmu_circ_0007131) knockdown most significantly impaired ILC3 differentiation (Fig. 1a). *CircTmem241* is generated by back-splicing of *Tmem241* transcript from exon 8 to exon 14 and conserved across various species (Supplementary Fig. 2a, b). Human ortholog *circTMEM241* was validated in human ILC3 by DNA sequencing (Supplementary Fig. 2c), indicating *circTmem241* was highly conserved between mice and humans. We observed that *circTmem241* was highly expressed in some tissues, especially in the bone marrow and digestive track (Fig. 1b). Among lymphocyte progenitors and immune cells tested, *circTmem241* was most highly expressed in CHILPs, ILCPs, and intestinal ILC3s (Fig. 1c), which was further confirmed by fluorescence in situ hybridization (FISH) assay (Supplementary Fig. 2d). In addition, *circTmem241* was mainly located in the nuclei of CHILPs, ILCPs, and ILC3s (Supplementary Fig. 2e).

To further determine the physiological role of *circTmem241* in the regulation of ILC3 commitment and function, we sought to generate *circTmem241*-deficient mice. The complementary elements flanking circRNA sequences are essential for their generation[21]. We screened out the complementary sequences in the introns flanking *circTmem241* and constructed plasmids for minigene assay (Supplementary Fig. 3a). We next generated *circTmem241* knockout (*circTmem241$^{-/-}$*) mice by deleting the downstream complementary element in the genome through CRISPR/Cas9 technology (Supplementary Fig. 3b). *CircTmem241* deletion was validated in *circTmem241$^{-/-}$* mice via PCR and qPCR analyses (Supplementary Fig. 3c, d). Of note, *circTmem241* deletion did not affect the expression of its maternal gene *Tmem241* (Supplementary Fig. 3d, e). We found that *circTmem241$^{-/-}$* mice displayed comparable numbers of ILC1s and ILC2s in isolated intestinal lamina propria lymphocytes compared to *circTmem241$^{+/+}$* littermate control mice (Fig. 1d, e). However, *circTmem241* deletion decreased intestinal ILC3 percentages and numbers (Fig. 1f, g), which was further verified by in situ immunofluorescence (IF) staining (Fig. 1h). Moreover, all three subsets of ILC3s displayed decreased numbers in *circTmem241*-deficient mice (Supplementary Fig. 3f). Consistently, *circTmem241*-deficient mice showed impaired development of gut-associated lymphoid tissues (Supplementary Fig. 3g). ILC3s are essential for gut homeostasis and bacterial defense via secreting IL-22[1]. As expected, *circTmem241* deletion decreased IL-22 positive ILC3s after *Citrobacter rodentium* infection and IL-22 production after IL-23 stimulation (Fig. 1i, j). Moreover, *circTmem241$^{-/-}$* mice were accompanied by greater feces bacteria load, more significant body weight loss, and severer Intestinal injury after *C. rodentium* infection than *circTmem241$^{+/+}$* littermates (Fig. 1k–n). Taken together, *circTmem241* is highly expressed in ILC3s and their progenitors and its deficiency reduces ILC3 numbers but not ILC1s or ILC2s.

### *CircTmem241* drives ILC3 specification from the ILCP progenitor stage

The reduced number of ILC3s might be caused by increased cell death and/or decreased development of cell populations. We then assessed the turnover rate of ILC3s in vivo. BrdU was intraperitoneally injected into mice at 100 mg/kg for 18 h, followed by detection of BrdU uptake by FACS. We found that ILC3s derived from *circTmem241$^{-/-}$* mice showed comparable BrdU uptake to those from littermate WT control mice (Supplementary Fig. 4a). Consistently, percentages of Ki67$^+$ ILC3s in *circTmem241$^{-/-}$* mice displayed similar levels compared with those of *circTmem241$^{+/+}$* mice (Supplementary Fig. 4b). In addition, *circTmem241* deficiency didn't affect death rates of ILC3s (Supplementary Fig. 4c), indicating that *circTmem241* does not affect cell proliferation and survival of ILC3s. To further determine the role of *circTmem241* deficiency on ILC3 reduction in vivo, we compared hematopoietic progenitor cells by FACS in *circTmem241$^{-/-}$* mice versus *circTmem241$^{+/+}$* littermate control mice. We noted that *circTmem241* deficiency did not affect the percentage and number of ILC progenitors, including CLPs, α$_4$β$_7^+$ CLPs, CHILPs, and ILCPs (Fig. 2a–d). ILCPs diversify into specific

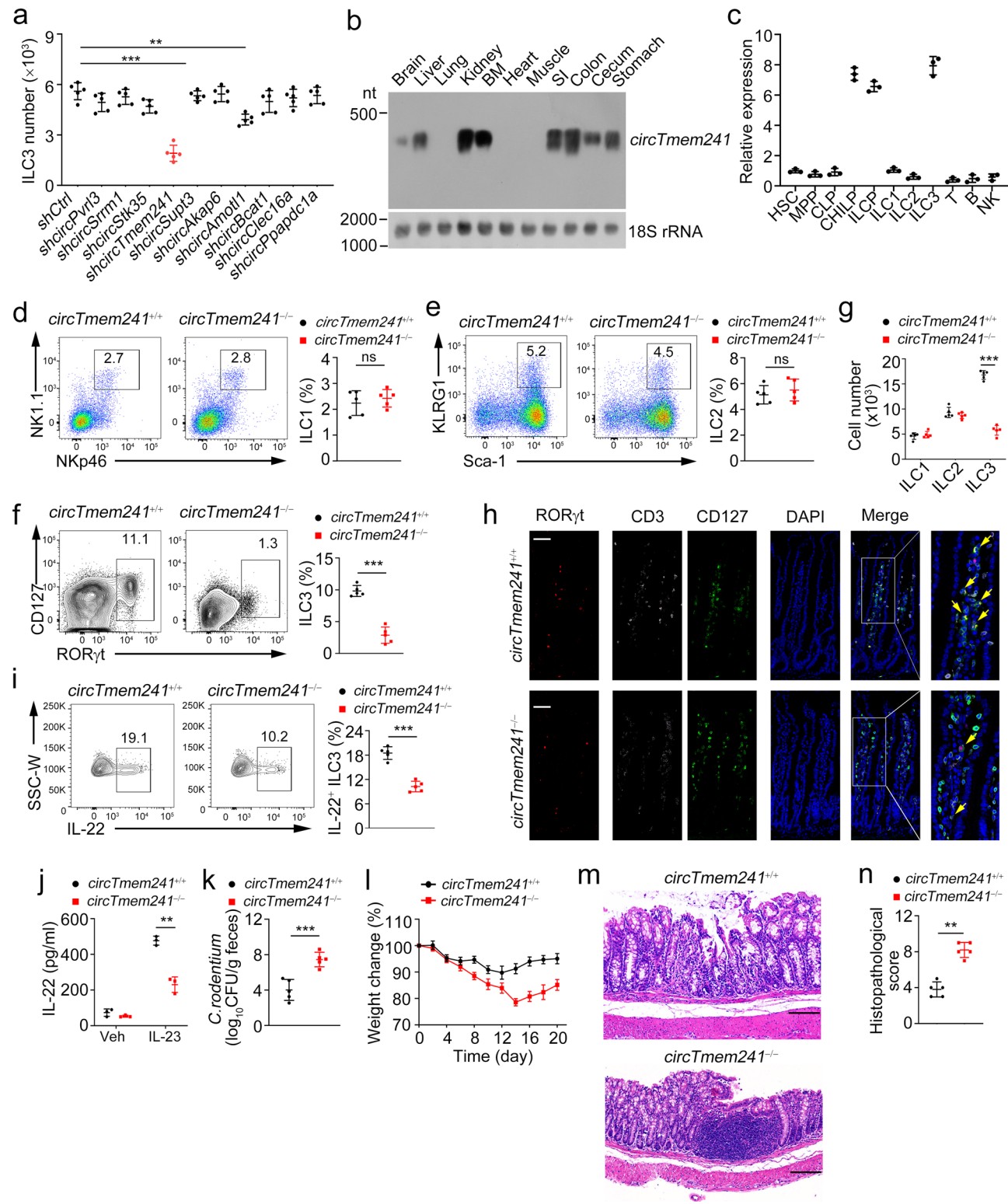

precursors of different ILC subsets before maturation[36,37]. We observed that *circTmem241* deficiency caused decreased percentages and numbers in ILC3Ps but not ILC1Ps or ILC2Ps (Fig. 2e–h). We next isolated CHILPs or ILCPs for in vitro differentiation assay and found that *circTmem241* knockout impaired ILC3P and ILC3 differentiation but not ILC1 or ILC2 (Fig. 2i, j and Supplementary Fig. 4d, e). Ectopic expression of *circTmem241* in *circTmem241*[−/−] CHILPs or ILCPs was able to rescue the differentiation ability of CHILPs or ILCPs toward ILC3Ps

and ILC3s (Fig. 2i, j). Altogether, *circTmem241* is required for ILC lineage commitment from ILCPs to ILC3Ps.

**CircTmem241 regulates ILC3 commitment in an intrinsic manner**
To further determine whether the role of *circTmem241* in ILC3 commitment is intrinsic or extrinsic, we conducted two groups of bone marrow transplantation experiments (Fig. 3a). For non-competitive bone marrow transplantation, we transplanted $5 \times 10^6$ CD45.2[+]

**Fig. 1 | *CircTmem241* knockout decreases ILC3 numbers. a** Top 10 highly expressed circRNAs were knocked down in CHILPs, followed by in vitro differentiation assay. ILC3s (CD3⁻CD19⁻CD127⁺CD45ˡᵒ RORγt⁺) were analyzed by FACS. $n = 5$ for each group. **b** *CircTmem241* expression levels of indicated tissues were detected by Northern blot using biotin-labeled probes. 18 S RNA was used as a loading control. **c** Relative expression of *circTmem241* was measured in indicated hematopoietic progenitors and mature cells by qPCR. Fold changes were normalized to endogenous *18 S*. HSC (Lin⁻c-Kit⁺Sca-1⁺CD150⁺CD48⁻), MPP (Lin⁻c-Kit⁺Sca-1⁺CD150⁻CD48⁺), CLPs (Lin⁻CD127⁺c-KitⁱⁿᵗSca-1ⁱⁿᵗFlt3⁺α4β7⁻), CHILPs (Lin⁻CD25⁻CD127⁺Flt3⁻α4β7⁺Id2ᴳᶠᴾ) and ILCPs (Lin⁻CD127⁺Flt3⁻c-Kit⁺α4β7⁺PLZFᴳᶠᴾ) were isolated from BM cells. ILC1s (CD3⁻CD19⁻CD45⁺CD127⁺NK1.1⁺NKp46⁺), ILC2s (Lin⁻CD127⁺CD90⁺KLRG1⁺Sca1⁺) and ILC3s (CD3⁻CD19⁻CD127⁺CD45ˡᵒ CD90ʰⁱ) were isolated from small intestinal lamina propria. T cells (CD3⁺CD45⁺) were isolated from thymus. B cells (NK1.1⁻CD19⁺CD45⁺) and NK cells (NK1.1⁺CD19⁻CD45⁺) were isolated from spleen. $n = 3$ biological independent experiments. **d–f** ILC1s (CD3⁻CD19⁻CD45⁺CD127⁺ gated), ILC2s (CD3⁻CD19⁻CD127⁺CD90⁺ gated), and ILC3s (CD3⁻CD19⁻CD45ˡᵒʷ gated) were analyzed in small intestines of *circTmem241⁺/⁺* and *circTmem241⁻/⁻* mice by FACS. $n = 5$ for each group. **g** Numbers of indicated ILCs in

*d–f* were calculated. $n = 5$ for each group. **h** Immunofluorescence staining of ILC3s in small intestines from *circTmem241⁺/⁺* and *circTmem241⁻/⁻* mice. Yellow arrows indicate ILC3s. Scale bar, 100 µm. **i** Lamina propria lymphocytes (LPLs) from *circTmem241⁺/⁺* and *circTmem241⁻/⁻* mouse intestines were sorted and stimulated by IL-23 for 4 h, followed by IL-22⁺ ILC3 detection with FACS. $n = 5$ for each group. **j** IL-22 levels secreted by ILC3s from *circTmem241⁺/⁺* and *circTmem241⁻/⁻* intestines were assayed by ELISA after stimulation by IL-23 for 24 h. $n = 3$ for each group. **k** Body weight changes of *circTmem241⁺/⁺* and *circTmem241⁻/⁻* mice were analyzed after *C. rodentium* infection. $n = 5$ for each group. **l** Bacterial counts (CFUs) in feces of *circTmem241⁺/⁺* and *circTmem241⁻/⁻* mice were measured after infection with *C. rodentium* for 6 days. $n = 5$ for each group. **m** Colon tissues from *circTmem241⁺/⁺* and *circTmem241⁻/⁻* mice were analyzed by H&E staining. Scale bars, 100 µm. **n** Histopathological scores of colons from *circTmem241⁺/⁺* and *circTmem241⁻/⁻* mice over *C. rodentium* infection. $n = 5$ for each group. **P < 0.01 and ***P < 0.001. Data were analyzed by an unpaired two-side Student's $t$ test and shown as means ± SD. Data are representative of at least three independent experiments. Source data are provided as a Source Data file.

*circTmem241⁻/⁻* or *circTmem241⁺/⁺* mouse BM cells into lethally irradiated CD45.1⁺ recipients. After 8 weeks, we found that recipients reconstituted with CD45.2⁺ *circTmem241⁻/⁻* BM cells decreased ILC3 numbers (Fig. 3b). For competitive bone marrow transplantation assays, we transferred 1:1 ratio mixture of CD45.1⁺ WT and CD45.2⁺ *circTmem241⁺/⁺* or *circTmem241⁻/⁻* BM cells into lethally irradiated recipient mice for 8 weeks. Recipients reconstituted with *circTmem241⁻/⁻* BM cells showed reduced ILC3 numbers compared to those reconstituted with *circTmem241⁺/⁺* BM cells (Fig. 3c). Consequently, *circTmem241⁻/⁻* BM cell reconstituted recipients were also susceptible to *C. rodentium* infection (Fig. 3d–f). These data indicate that *circTmem241* regulates ILC3 commitment and anti-bacterial immunity in an intrinsic manner.

## *CircTmem241* promotes *Elk3* expression

To explore the molecular mechanism of *circTmem241* in the regulation of ILC3 commitment, we crossed *circTmem241⁻/⁻* mice with *PLZF*ᴳᶠᴾ mice and isolated ILCPs from *PLZF*ᴳᶠᴾ or *PLZF*ᴳᶠᴾ*circTmem241⁻/⁻* mice for transcriptome analysis. We found that *circTmem241* deletion caused downregulation of many TFs (Fig. 4a). Among the top 10 downregulated TFs, *Elk3* was most downregulated (Fig. 4b). Furthermore, knockdown of *Elk3* significantly decreased percentages of ILC3s by in vitro differentiation assay (Fig. 4c and Supplementary Fig. 4f). *Elk3* downregulation in *circTmem241⁻/⁻* ILCPs was also confirmed by Western blot (Fig. 4d). Elk3 overexpression could rescue the reduction of ILC3s by in vitro differentiation assay, indicating Elk3 played a critical role in *circTmem241*-mediated ILC3 commitment (Supplementary Fig. 4g, h). Through chromatin isolation by RNA purification (CHIRP) assay, we observed that *circTmem241* was enriched on the promoter region (−200-0) of *Elk3* (Fig. 4e). We also found that *circTmem241* bound to the segment (−200-0) of *Elk3* promoter via hybridization with biotinylated *circTmem241* RNA probe (Fig. 4f). This result was further validated by luciferase assay (Fig. 4g). These data suggest that *circTmem241* binds to *Elk3* promoter to regulate its transcription.

Previous studies have revealed that circRNA loops are involved in the interaction of circular RNAs[20]. We predicted the secondary structure of *circTmem241* with a bioinformatics tool and identified four loop regions of *circTmem241* transcript (Supplementary Fig. 5a). To determine which loop (HR1 (hairpin region #1, HR1), HR2, HR3, and HR4) was required for the association between *circTmem241* transcript and *Elk3* promoter, we constructed several *circTmem241* mutations to abrogate loop structures (Supplementary Table 1). We found that HR1 was required for their interaction (Fig. 4h). Through sequence alignment, we noticed that pairing complementary bases existed in the *Elk3* promoter (−200-0) versus the HR1 of *circTmem241* (Supplementary Fig. 5b). We generated the mutations of *circTmem241* transcript to delete their base pairing bases (*circTmem241*-mut), followed by a

hybridization assay. We found that *circTmem241*-mut really abolished the interaction between *circTmem241* transcript and *Elk3* promoter (Fig. 4i). In addition, substitution mutation in *Elk3* promoter pairing region failed to activate luciferase activity by *circTmem241* overexpression (Supplementary Fig. 5c). Through DNase I accessibility assay, we found that *Elk3* promoter in *circTmem241⁻/⁻* ILCPs was less susceptible to DNase I digestion (Fig. 4j). Consistently, *circTmem241* deficiency caused attenuated transcription of *Elk3* mRNA by nuclear run-on assay (Fig. 4k). To further confirm the important role of the pairing bases, we conducted in vitro differentiation assay using ILCPs from *circTmem241⁺/⁺* or *circTmem241⁻/⁻* mice. We observed that only overexpression of *circTmem241*-WT was able to rescue ILC3 differentiation (Fig. 4l). Altogether, *circTmem241* binds to *Elk3* promoter to promote its expression, leading to the commitment of ILC3s.

## *CircTmem241* associates with Nono in ILCPs

To further explore how *circTmem241* regulates *Elk3* transcription in ILCPs, we conducted RNA-pulldown assay using biotin-labeled *circTmem241* transcripts to pulldown their interacting components from bone marrow cells, followed by sodium dodecyl sulfate−polyacrylamide gel electrophoresis (SDS-PAGE), silver staining, and mass spectrometry. We found that *circTmem241* bound to Nono from bone marrow lysates (Fig. 5a and Supplementary Fig. 5d). We further validated the interaction of *circTmem241* with Nono via CHIRP assay and immunoblotting analysis (Fig. 5b, c). In addition, *circTmem241* was enriched by anti-Nono antibody through RNA immunoprecipitation assay (Fig. 5d). Moreover, the interaction between *circTmem241* and Nono was further validated by electrophoretic mobility shift assay (EMSA) assay (Fig. 5e). Consistently, *circTmem241* was co-localized with Nono in ILCP cells (Fig. 5f).

To further determine which region of *circTmem241* transcript was required for its interaction with Nono, we performed RNA-pulldown assay using biotin-labeled truncated *circTmem241* transcripts. We observed that the 161−240 region truncation of *circTmem241* transcript (Δ161−240) abolished the association between *circTmem241* and Nono (Fig. 5g). In addition, the N-terminal RNA recognition motif (RRM) of Nono protein was responsible for its interaction with *circTmem241* (Fig. 5h). Nono overexpression significantly enhanced luciferase activity of *Elk3* promoter (Fig. 5i). Moreover, *Elk3* transcription was remarkably attenuated after Nono knockout (Fig. 5j and Supplementary Fig. 5e, f). Collectively, *circTmem241* directly interacts with Nono protein in ILCPs.

## *CircTmen241* recruits histone methyltransferase Ash1l onto *Elk3* promoter to trigger its transcription

We next sought to explore how *circTmem241* regulated *Elk3* transcription. We used anti-Nono antibody to go through bone marrow cell lysates for immunoprecipitation assay. Precipitated candidates were

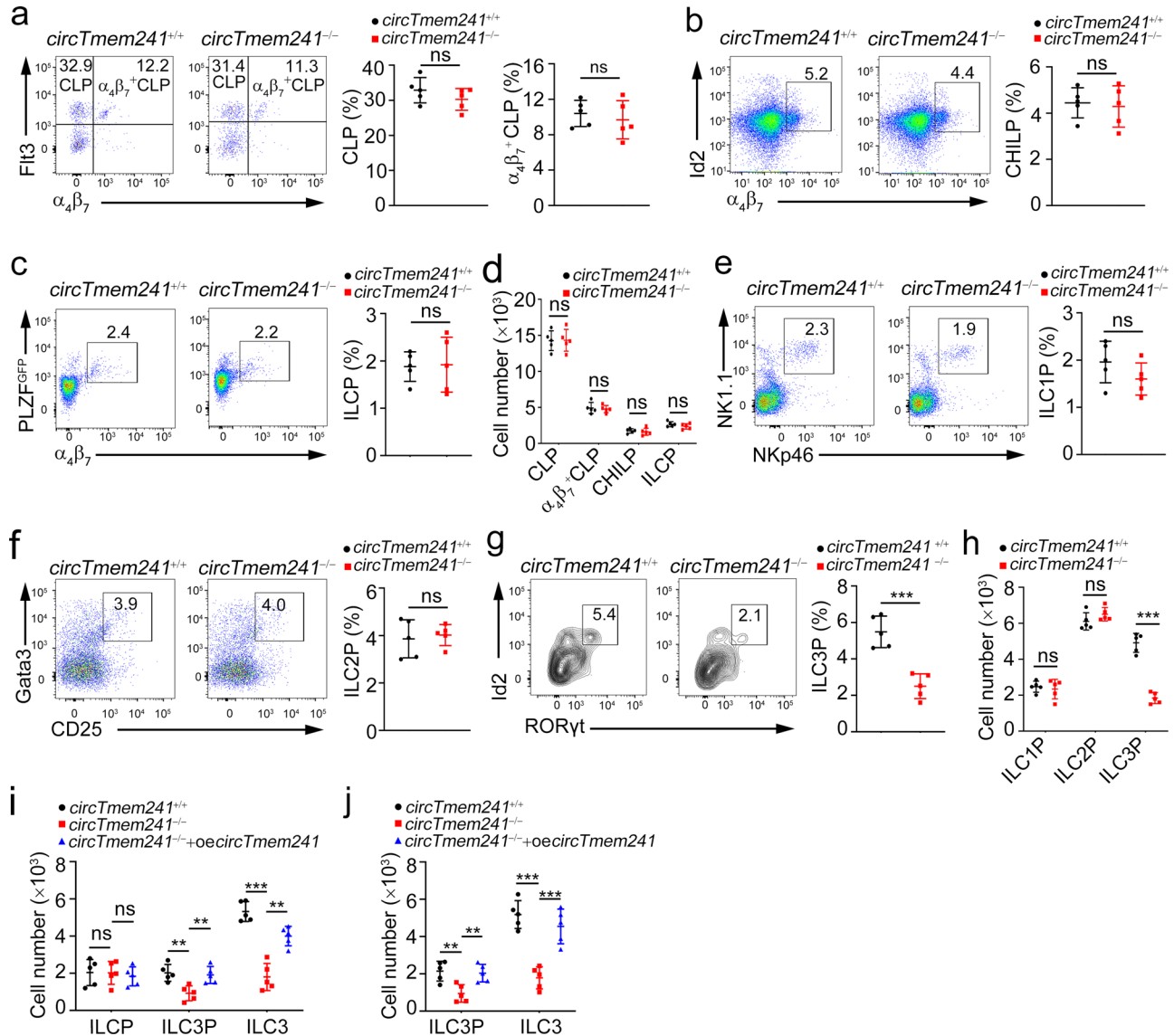

**Fig. 2 | *CircTmem241* drives ILC3 differentiation from the ILCP progenitor stage. a–c** FACS analysis of *circTmem241*$^{+/+}$ and *circTmem241*$^{-/-}$ ILC lineage progenitors. Plots displayed frequencies of CLPs (Lin⁻CD127⁺c-Kit$^{int}$Sca-1$^{int}$ gated), α₄β₇⁺CLPs (Lin⁻CD127⁺c-Kit$^{int}$Sca-1$^{int}$ gated), CHILPs (Lin⁻CD25⁻CD127⁺Flt3⁻ gated) and ILCPs (Lin⁻CD127⁺Flt3⁻c-Kit⁺ gated). *n* = 5 for each group. ns, no significance. **d** Numbers of indicated progenitor cells in **a–c** were calculated. *n* = 5 for each group. ns, no significance. **e–g** Precursor cells ILC1P (Lin⁻CD127⁺Eomes⁻CD49a⁺ gated), ILC2P (Lin⁻CD127⁺CD45⁺Flt3⁻CD117⁻Sca-1⁺ gated) and ILC3P (Lin⁻CD25⁻CD127⁺α4β7$^{int}$Gata3$^{low}$Bcl11b⁺ gated) were analyzed in *circTmem241*$^{+/+}$

and *circTmem241*$^{-/-}$ mice by FACS. *n* = 5 for each group. **h** Numbers of indicated ILC precursor cells in **e–g** were calculated. *n* = 5 for each group. ns, no significance. **i–j** CHILPs (Lin⁻CD25⁻CD127⁺Flt3⁻α4β7⁺) or ILCPs (Lin⁻CD25⁻CD127⁺Flt3⁻α4β7⁺c-Kit⁺PD-1⁺) were sorted from *circTmem241*$^{+/+}$ and *circTmem241*$^{-/-}$ mice and cultured under in vitro differentiation condition for 7 or 14 days. Flow cytometric analysis of indicated cells was performed and absolute numbers were calculated. *n* = 5 for each group. **P < 0.01 and ***P < 0.001. Data were analyzed by an unpaired two-side Student's *t* test and shown as means ± SD. Data are representative of at least three independent experiments. Source data are provided as a Source Data file.

separated by SDS-PAGE, followed by silver staining and mass spectrometry. A major differential band was identified to be the histone methyltransferase Ash1l (Fig. 6a and Supplementary Fig. 6a). The interaction of Nono with Ash1l was validated by co-IP assay (Fig. 6b). By RNA-pulldown assay, *circTmem241* failed to pulldown Ash1l from Nono deleted bone marrow lysates (Fig. 6c). Consistently, *circTmem241* didn't interact with Ash1l directly by in vitro binding assay (Fig. 6d), suggesting that Nono acted as an adaptor protein for the association of *circTmem241* with Ash1l. Of note, Nono co-localized with *Elk3* promoter in WT ILCPs but not in *circTmem241*$^{-/-}$ ILCPs (Fig. 6e). Moreover, with cross-linking treatment, Ash1l was co-eluted with *Elk3* promoter in *circTmem241*$^{+/+}$ bone marrow cell lysates, but not in *circTmem241*$^{-/-}$ cell lysates (Fig. 6f).

Ash1l is mainly responsible for H3K4me3 and H3K36me3 modifications that promote chromatin accessibility for gene transcription[38]. Through Chromatin immunoprecipitation (ChIP) assay with indicated antibodies, Ash1l was enriched on *Elk3* promoter region (−200-0) and *circTmem241* knockout decreased H3K4me3 and H3K36me3 modification levels (Fig. 6g–i). These data suggest that *circTmem241* recruits Ash1l onto *Elk3* promoter that facilitates its histone methylation modifications in ILCPs. To further test whether Nono and Ash1l regulated *Elk3* transcription, we deleted *Ash1l* via AAV delivery of guide RNAs in ILCPs from *Vav-Cre;Cas9-KI* mice (Supplementary Fig. 6b, c). We found that *Elk3* promoter was less enriched with H3K4me3 and H3K36me3 modifications in *Nono*$^{-/-}$ and *Nono*$^{-/-}$*circTmem241*$^{-/-}$ ILCPs (Fig. 6j and Supplementary Fig. 6d). In addition, *Elk3* mRNA was less transcribed in *Nono*$^{-/-}$ and *Nono*

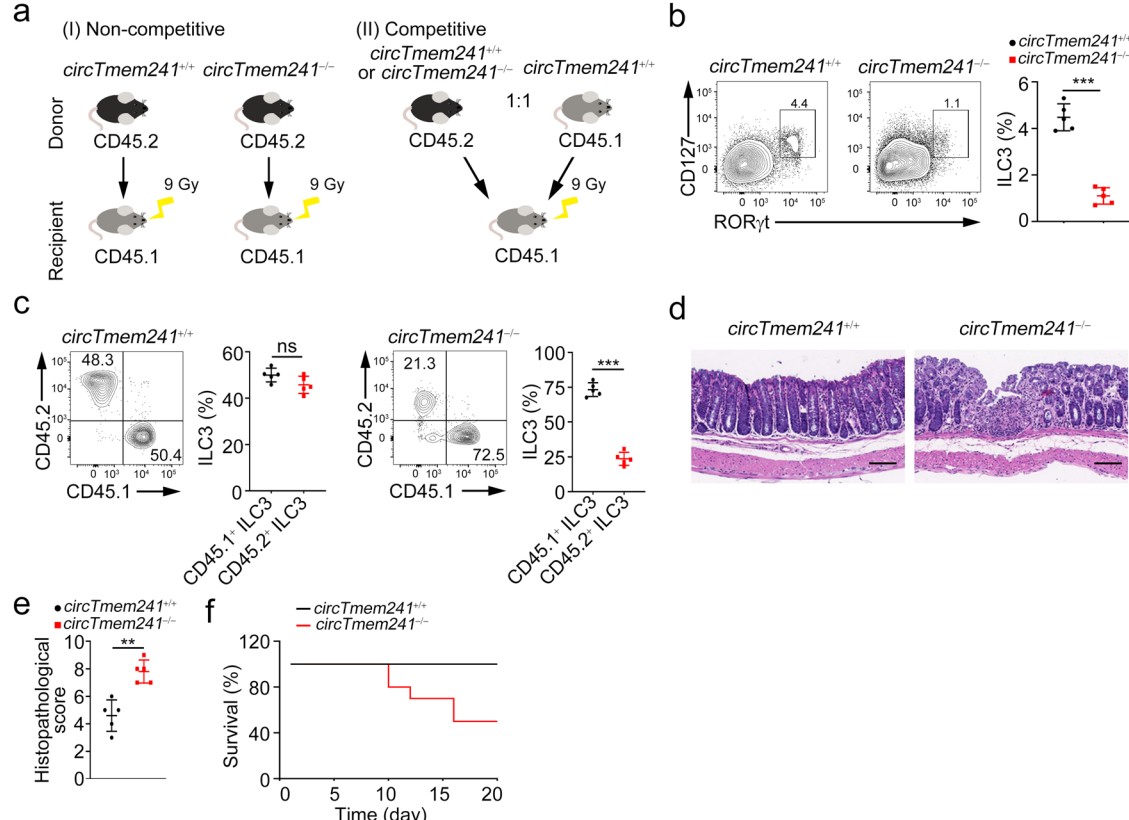

**Fig. 3 | *CircTmem241* regulates ILC3 commitment in an intrinsic manner.**
**a** Schematic diagram of bone marrow transplantation. **b** $5 \times 10^6$ CD45.2$^+$ BM cells from *circTmem241*$^{+/+}$ or *circTmem241*$^{-/-}$ mice were transferred into lethally irradiated CD45.1$^+$ recipients. 8 weeks later, ILC3s (Lin$^-$CD45.2$^+$CD127$^+$RORγt$^+$) were analyzed. $n = 5$ for each group. **c** 1:1 mixture of CD45.1$^+$ wild-type and CD45.2$^+$ *circTmem241*$^{+/+}$ or *circTmem241*$^{-/-}$ BM cells were transferred into lethally irradiated CD45.1$^+$ recipient mice for 8 weeks. Then ratios of CD45.1$^+$ to CD45.2$^+$ ILC3s in chimeras were analyzed by gating on CD45.2$^+$Lin$^-$RORγt$^+$ (*circTmem241*$^{+/+}$ or *circTmem241*$^{-/-}$) and CD45.1$^+$Lin$^-$RORγt$^+$ (WT). $n = 5$ for each group. **d** Irradiated

recipient mice were reconstructed with *circTmem241*$^{+/+}$ or *circTmem241*$^{-/-}$ BM cells as indicated in **a** (left). Colons isolated from reconstructed mice infected with *C. rodentium* were subjected to HE staining. Scale bars, 100 μm. **e** Histological scores of colons from reconstructed mice in **d**. $n = 5$ for each group. **f** Reconstituted mice, as indicated in **d**, were infected with *C. rodentium* and survival rates were measured. $N = 10$ for each group. **\*\***$P < 0.01$ and **\*\*\***$P < 0.001$. Data were analyzed by an unpaired two-side Student's *t* test and shown as means ± SD. Data are representative of at least three independent experiments. Source data are provided as a Source Data file.

$^{-/-}$*circTmem241*$^{-/-}$ ILCP cells by nuclear run-on assay (Fig. 6k). Consequently, Elk3 expression was remarkably inhibited in *Nono*$^{-/-}$ and *Nono*$^{-/-}$*circTmem241*$^{-/-}$ ILCPs (Fig. 6l, m). In parallel, Ash1l knockout also exhibited less enriched histone modifications and dramatically attenuated Elk3 expression (Fig. 6j–m and Supplementary Fig. 6d). Taken together, *circTmem241* recruits the histone methyltransferase Ash1l to initiate *Elk3* transcription in a Nono-dependent manner.

### *Elk3* deficiency impairs ILC3 specification and anti-bacterial immunity

We next established *Elk3*-deficient mice via a CRISPR/Cas9 approach (Supplementary Fig. 6e–g). We performed a series of in vitro differentiation assays to verify the *circTmem241*-Nono-Ash1l-Elk3 axis in ILC3 commitment. We found that *Elk3*$^{-/-}$ ILCPs showed decreased ILC3P and ILC3 numbers (Fig. 7a, b). By contrast, Elk3 overexpression in *Elk3*$^{-/-}$ ILCPs was able to rescue ILC3 numbers (Fig. 7b). Of note, deletion of *Nono* or *Ash1l* in ILCPs suppressed the differentiation of ILC3s, which could be rescued by *Nono*, *Ash1l* or *Elk3* overexpression (Fig. 7c). Moreover, ILCPs of *circTmem241*$^{-/-}$ or *circTmem241*$^{-/-}$ *Elk3*$^{-/-}$ mice showed significantly attenuated ability to differentiate into ILC3s, which could be partially rescued by Elk3 overexpression (Fig. 7d). Notably, *circTmem241* overexpression in *circTmem241*$^{-/-}$*Elk3*$^{-/-}$ ILCPs failed to restore ILC3 numbers, while overexpression of both *circTmem241* and Elk3 almost fully rescued ILC3 differentiation ability (Fig. 7d). Taken together, the *circTmem241*-Nono-Ash1l-Elk3 axis is required for ILC3 differentiation.

As expected, *Elk3* deletion markedly decreased ILC3 numbers but did not affect the numbers of ILC1s or ILC2s in vivo (Fig. 7e–h). In addition, the development of gut-associated lymphoid tissues was also impaired in *Elk3*-deficient mice (Supplementary Fig. 6h). Consistently, numbers of IL-22$^+$ ILC3s were also dramatically reduced in *Elk3*$^{-/-}$ mice (Fig. 7i). Importantly, *circTmem241*$^{-/-}$*Elk3*$^{-/-}$ mice displayed much lower numbers of ILCPs and ILC3s (Fig. 7j) as well as were much more susceptible to bacterial infection than their littermate WT or *Elk3*$^{-/-}$ mice (Fig. 7k–m). Collectively, *Elk3* regulates *circTmem241*-mediated ILC3 differentiation that exerts anti-bacterial immunity.

### Discussion

ILC3s play important roles in the early innate immune response against pathogens and maintaining tissue homeostasis[4]. Abnormality of ILC3 development causes severe intestinal diseases after infection. Here we identified an undefined circular RNA *circTmem241* that is highly expressed in ILC3s and their progenitor cells. *CircTmem241* deficiency impairs ILC3 commitment and consequently attenuates anti-bacterial immunity against *C. rodentium* infection. Mechanistically, *circTmem241* interacts with Nono protein to recruit histone methyltransferase Ash1l onto *Elk3* promoter in ILCP cells. Ash1l-mediated histone modifications on *Elk3* promoter enhance chromatin accessibility to initiate *Elk3* transcription. Of note, *circTmem241*$^{-/-}$, *Nono*$^{-/-}$, and *Ash1l*$^{-/-}$ ILCPs display impaired ILC3 differentiation, while Elk3 overexpression rescues ILC3 commitment ability. Finally, *circTmem241*$^{-/-}$*Elk3*$^{-/-}$ mice show much lower numbers of ILC3s and are much more susceptible to

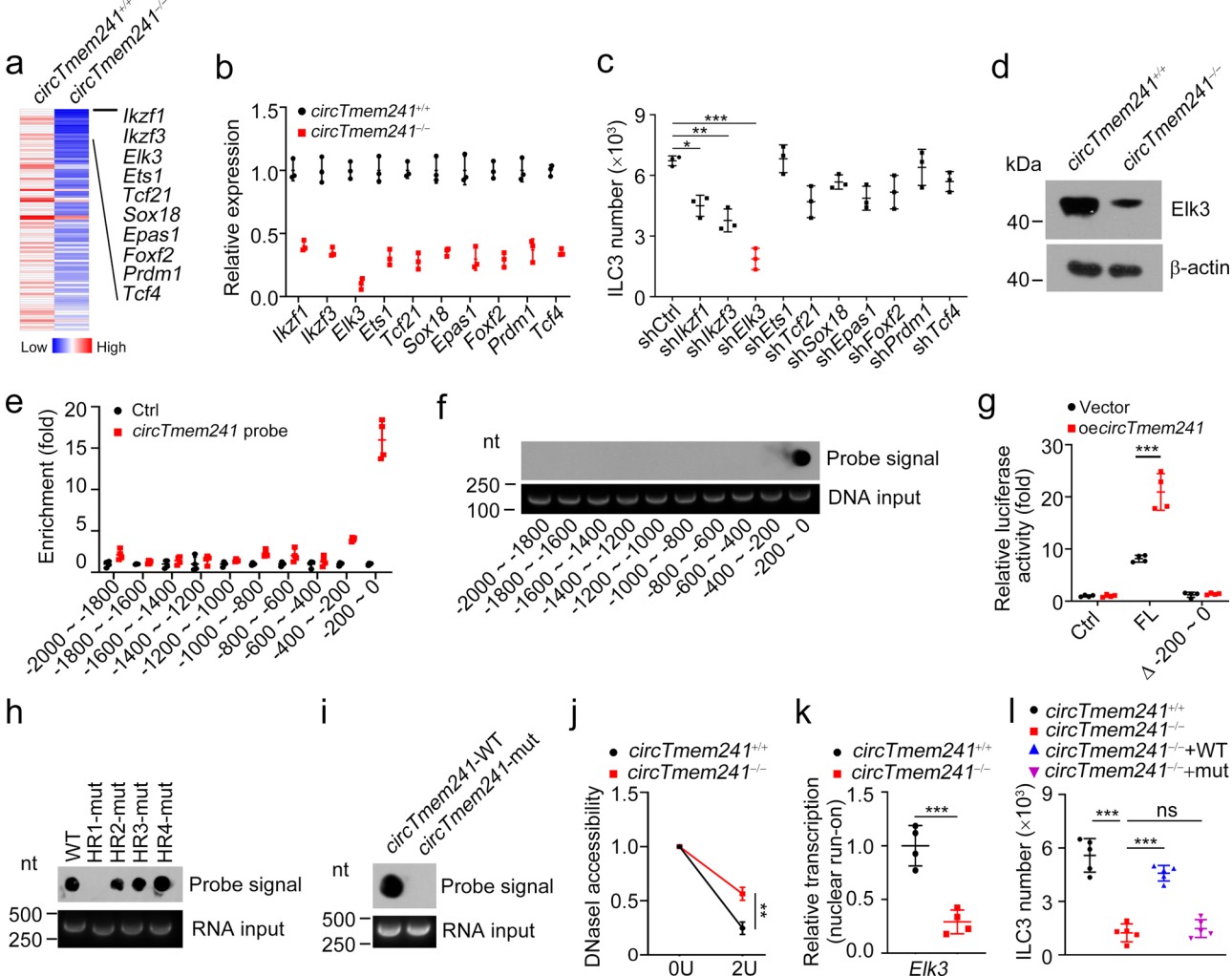

**Fig. 4 | *CircTmem241* enhances *Elk3* expression. a** Heatmap of top ten down-regulated TFs in *circTmem241*[−/−] ILCPs versus *circTmem241*[+/+] ILCPs by transcriptome analysis. **b** Relative expression levels of top 10 TFs in *circTmem241*[+/+] and *circTmem241*[−/−] ILCPs were analyzed by qRT-PCR. Fold changes were normalized to endogenous *Gapdh*. *n* = 3 biological independent experiments. **c** Top 10 down-regulated TFs were knocked down in ILCPs were analyzed by FACS. *n* = 3 biological independent experiments. **d** Elk3 protein levels in *circTmem241*[+/+] or *circTmem241*[−/−] ILCPs by western blotting. **e** Enrichment of *circTmem241* on *Elk3* gene promoter was analyzed by CHIRP assay. *CircTmem241* probe was biotin labeled. *n* = 4 for each group. **f** Enrichment of *circTmem241* on *Elk3* gene promoter was further validated by dot blot. **g** Luciferase reporter assay was performed to validate *circTmem241* function on *Elk3* transcription activation. FL, full-length (represent −2000-0 region upstream *Elk3* transcriptional start site). *n* = 4 for each group. **h**, **i** *CircTmem241*

hairpin region (HR) sequence mutations were incubated with *Elk3* promoter regions, followed by dot blot. **j** DNase I assay of chromatin accessibility in *Elk3* promoters of *circTmem241*[+/+] or *circTmem241*[−/−] ILCPs by qRT-PCR. *n* = 3 for each group. **k** Transcription activities of *Elk3* in *circTmem241*[+/+] or *circTmem241*[−/−] ILCPs were measured by nuclear run-on assay. *n* = 4 for each group. **l** ILCPs from *circTmem241*[+/+] or *circTmem241*[−/−] mice were infected with virus expressing *circTmem241*-WT or *circTmem241*-mut, followed by in vitro differentiation assay. ILC3s (CD3⁻CD19⁻CD127⁺CD45[lo]RORγt⁺) were analyzed by FACS. *n* = 5 for each group. WT, wild-type *circTmem241*; mut, *circTmem241* mutation. **P* < 0.05, ***P* < 0.01, and ****P* < 0.001. Data were analyzed by an unpaired two-side Student's *t* test and shown as means ± SD. Data are representative of at least three independent experiments. Source data are provided as a Source Data file.

bacterial infection. Therefore, we conclude that the *circTmem241*-Nono-Ash1l-Elk3 axis is required for ILC3 differentiation.

Many circRNAs contain miRNA-binding elements that act as miRNA sponges to exert their functions[20]. Instead, circRNAs without miRNA sponge potential utilize alternative mechanisms to exert their roles. Some circRNAs work as scaffolds for their interactions with proteins, DNAs, RNAs, or other molecules[26]. For instance, circBIRC6 interacts with AGO2 to maintain stem cell pluripotency[39]. We previously demonstrated that *circKcnt2* associates with Mbd3 to inhibit Batf transcription, leading to ILC3 inactivation[27]. *CircPan3* binds Il13ra1 mRNA in the intestinal stem cells (ISCs) to increase its stability, which results in the expression of IL-13Rα1 in ISCs to sustain their self-renewal[40]. *cia-cGAS* binds DNA sensor cGAS in the nucleus of HSCs to block its synthase activity, thereby protecting dormant HSCs from

cGAS-mediated exhaustion[27]. *CircTmem241* is derived of *Tmem241* transcripts from exon 8 to exon 14, whose formation depends on upstream and downstream complementary sequences of the *Tmem241* transcripts. We show that *circTmem241* is highly expressed in ILC progenitors and regulates ILC3 specification from the ILCP stage. *CircTmem241* knockout does not affect the mRNA levels of its parental gene *Tmem241* and other neighbor genes. *CircTmem241* associates with Nono protein to recruit Ash1l to initiate *Elk3* transcription, which drives ILC3 specification from the ILCP stage. We also found that *circTmem241* was highly expressed in the intestine and mature ILC3s. Tmem241 is a predicted sugar transport protein, raising the possibility of nutrient component regulation on *circTmem241* formation and function of ILC3s. Previous studies showed that ILC3s-mediated immune response can be modulated by nutrients-derived

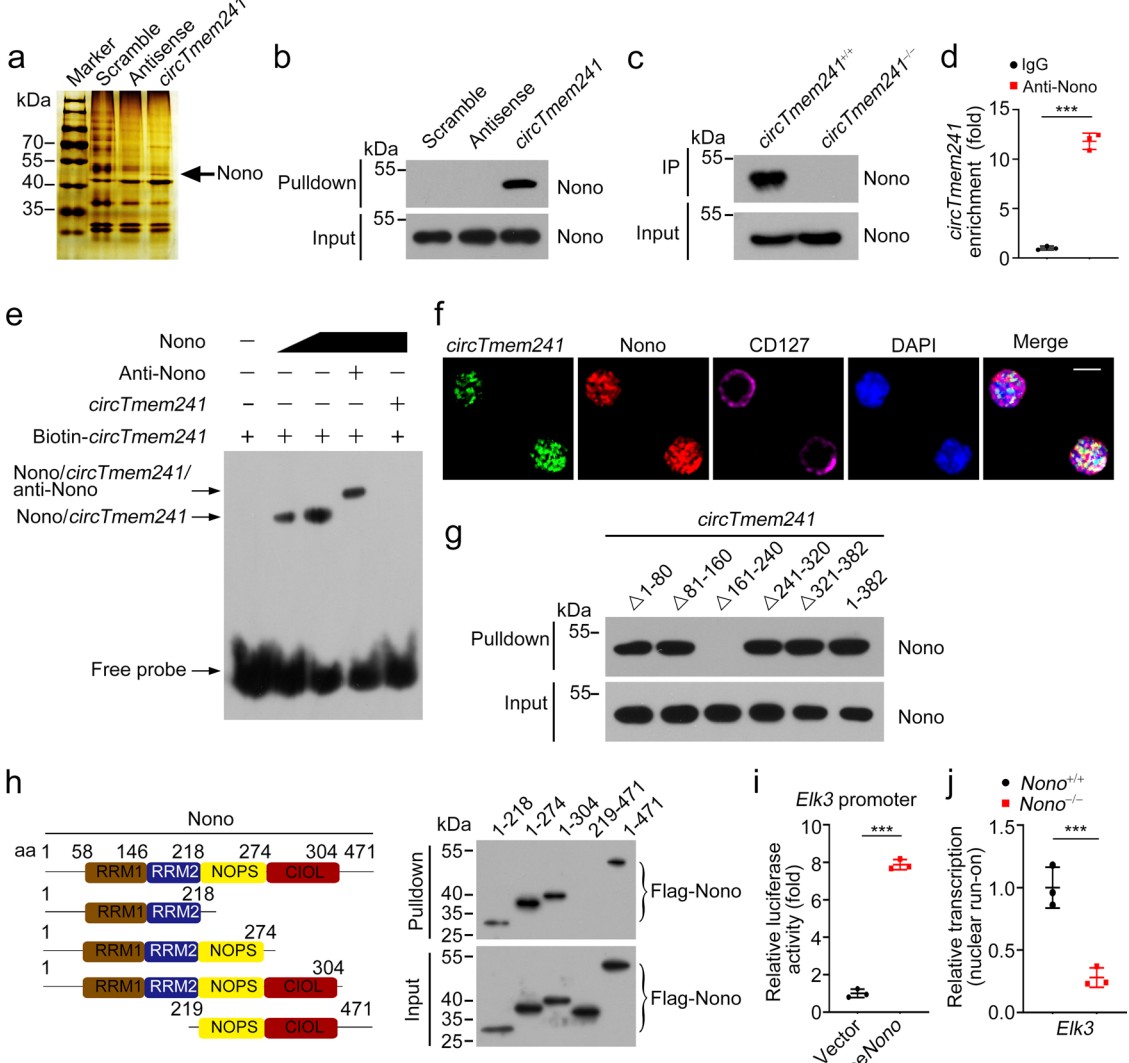

**Fig. 5 | *CircTmem241* directly interacts with Nono. a** BM cells from WT mice were lysed and incubated with in vitro transcribed linear biotin-labeled *circTmem241* transcripts, antisense, or scramble control. Pulldown components were separated by SDS-PAGE and followed by silver staining. Binding proteins of *circTmem241* were identified via mass spectrometry. **b** BM cell lysates were incubated with *circTmem241* probes or controls, followed by Western blotting. **c** Association between *circTmem241* and Nono was validated by CHIRP assay, followed by immunoblotting. **d** RNA immunoprecipitation (RIP) assay was conducted using anti-Nono antibody or IgG through BM cell lysates and *circTmem241* enrichment was detected by qRT-PCR. *n* = 3 for each group. **e** EMSA was performed using biotin-labeled *circTmem241* transcripts and recombinant Nono proteins with or without anti-Nono antibody. **f** *CircTmem241* was colocalized with Nono in ILCPs by immunofluorescence staining

and confocal imaging. Scale bar, 10 μm. **g** RNA-pulldown assay was performed using indicated truncations of linear biotin-labeled *circTmem241* RNAs with BM cell lysates. Interaction of Nono with indicated *circTmem241* mutants was analyzed by western blotting. **h** Validation of binding ability of *circTmem241* with indicated Nono truncations by RNA-pulldown assay, followed by western blotting. **i** Luciferase reporter assay was conducted to validate Nono function on *Elk3* transcription activation. *n* = 3 for each group. **j** Transcription activity of *Elk3* in *Nono*[+/+] or *Nono*[-/-] ILCPs was measured by nuclear run-on assay. *n* = 3 for each group. ***P* < 0.001. Data were analyzed by an unpaired two-side Student's *t* test and shown as means ± SD. Data are representative of at least three independent experiments. Source data are provided as a Source Data file.

microenvironments[41–43]. For instance, Vitamin A deficiency results in impaired ILC3 effector function and dramatic ILC2 amplification, which represents an adaption of local immune system against ongoing mucosal barrier challenges[42]. However, the formation and decay of *circTmem241* in ILCPs and ILC3s needs to be further investigated.

Cell fate decision is a sophisticated process that is strictly regulated by both genetic and epigenetic factors[44,45]. Histone modifications remodel chromatin status to modulate global genome accessibility for gene transcription initiation. Different kinds of histone modifications (called 'histone code'), including methylation, acetylation, phosphorylation, and ubiquitylation, work separately or together to regulate gene expression[46,47]. Accumulating evidence shows that epigenetic modifications play important roles

in ILC commitment and functions[6,48,49]. We previously showed that Yeats4 drives ILC lineage commitment via recruitment of Dot1l to activate *Lmo4* expression[50]. We also showed that lnckdm2b recruits Satb1 and the NURF complex to initiate Zfp292 transcription, contributing to the maintenance of ILC3 homeostasis[18]. In this study, we showed that *circTmem241* directly interacts with Nono to recruit Ash1l onto *Elk3* promoter in ILCPs. Ash1l as a histone methyltransferase catalyzes histone methylation modifications at H3K4 and H3K36 sites, which is tightly associated with active gene transcription[38]. Ash1l contributes to the enrichment of H3K4me3 and H3K36me3 on *Elk3* promoter, which triggers *Elk3* transcription to drive ILC3 commitment. Thus, *circTmem241*-mediated ILC3 commitment is in an epigenetic manner.

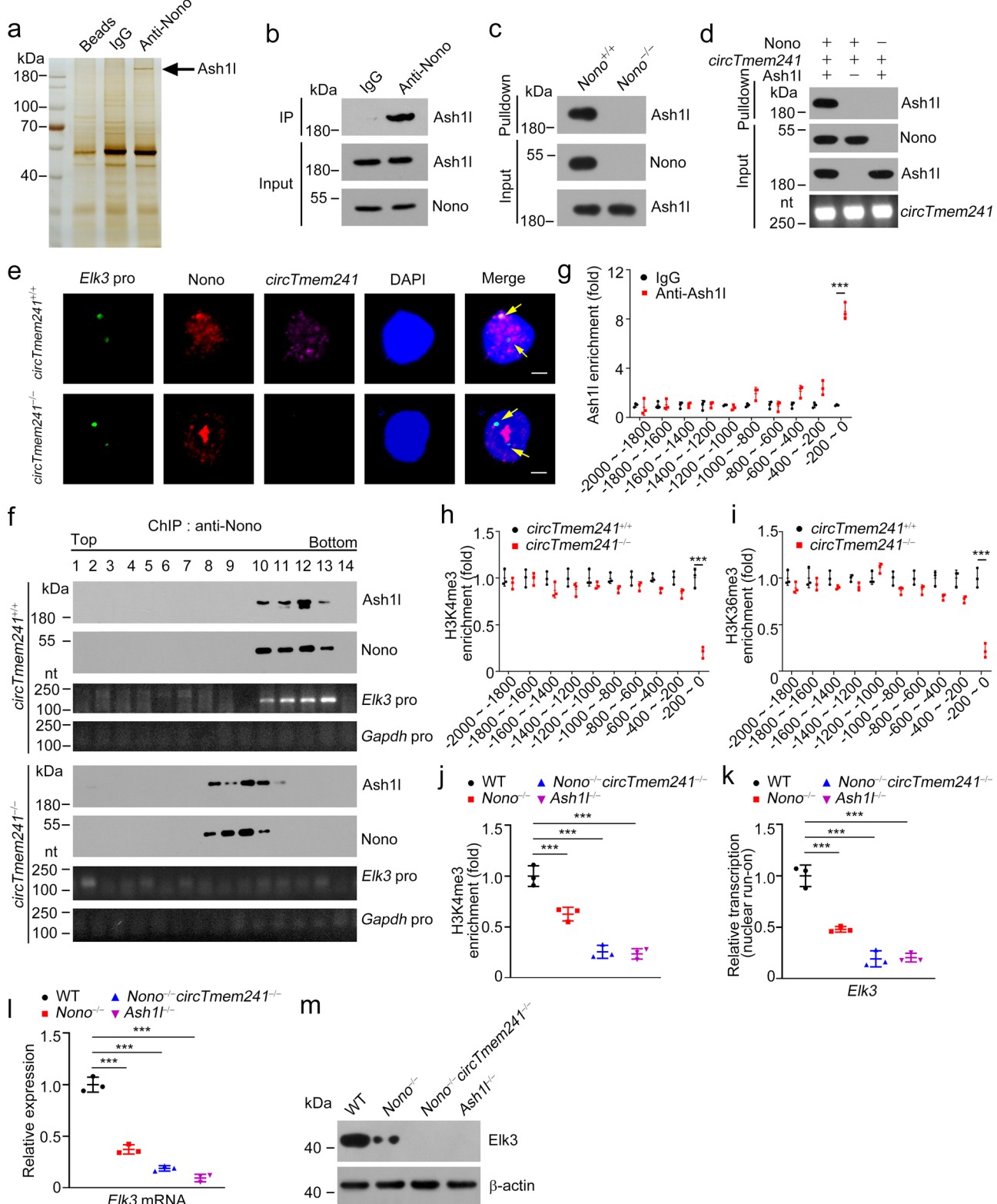

ILC specification is a multi-step and finely controlled process. Several ILC progenitors, including EILPs, CHILPs, and ILCPs, have been identified and their major transcription factors regulating their downstream differentiation have also been defined. ILCPs are characterized by the expression of transcription factor PLZF, which drives differentiation of all groups of ILCs[51]. However, how the precursors downstream of ILCPs generate is still elusive. A recent scRNA-seq study defined several new subpopulations of ILC

progenitors in bone marrow[14]. They clustered one subpopulation as the ILC3P cells that harbor the potential to specifically differentiate into ILC3s. However, how ILCPs differentiate into ILC3Ps and ILC3s remains unclear. Herein we identified an undefined circular RNA *circTmem241* highly expressed in ILCPs that drives ILC3 specification from the ILCP to ILC3P stage. We reveal that the *circTmem241*-Nono-Ash1l-Elk3 axis is required for the ILCP differentiation into ILC3P and ILC3 maturation, which is important to

**Fig. 6 | *CircTmem241* recruits Ash1l onto *Elk3* promoter to initiate its transcription. a** Immunoprecipitation assay was performed using BM cells from WT mice with anti-Nono antibody or IgG. Eluted fractions were resolved by SDS-PAGE, followed by silver staining and mass spectrometry. **b** BM cell lysates were incubated with anti-Nono antibody or IgG, followed by western blotting. **c** RNA-pulldown assay using biotin-labeled *circTmem241* transcripts with lysates from *Nono*$^{+/+}$ and *Nono*$^{-/-}$ bone marrow cells. **d** Interaction between *circTmem241* and Ash1l was measured by in vitro binding assay. **e** DNA FISH showed that *Elk3* promoter co-localized with Nono and *circTmem241*. Scale bar, 2 μm. **f** BM cells from *circTmem241*$^{+/+}$ and *circTmem241*$^{-/-}$ mice were lysed and treated with 1% formaldehyde for cross-linking. Anti-Nono antibody was incubated with treated lysates for ChIP assays, followed by size fractionation with sucrose gradient ultracentrifugation. Eluate gradients were examined by Western blotting and PCR assay. **g** Enrichment of Ash1l on *Elk3* gene promoter was analyzed by ChIP assay with anti-Ash1l antibody. *n* = 3 for each group. **h, i** Enrichment of indicated histone modifications on *Elk3* promoter was analyzed by ChIP assay. *n* = 3 for each group. **j** Enrichment of H3K4me3 on the *Elk3* promoter in indicated ILC progenitor cells (Lin$^-$CD127$^+$c-Kit$^{int}$Sca-1$^{int}$α$_4$β$_7^+$) was examined. *n* = 3 for each group. **k** WT, *Nono*-deficient, or *Nono*- and *circTmem241*-deficient ILCPs were subjected to nuclear run-on assay, followed by RT-PCR analysis. *n* = 3 for each group. **l, m** Relative mRNA **l** and protein **m** levels of *Elk3* in indicated ILCPs were analyzed. *n* = 3 for each group in **l**. ***$P$ < 0.001. Data were analyzed by an unpaired two-side Student's *t* test and shown as means ± SD. Data are representative of at least three independent experiments. Source data are provided as a Source Data file.

manipulate this axis for ILC development on treatment of infectious diseases.

# Methods

All experiments in this article comply with all relevant ethical regulations and are approved by the Institutional Committee of Institute of Biophysics, Chinese Academy of Sciences. The animal experimental protocols were approved by the Institutional Animal Care and Use Committee of Institute of Biophysics, Chinese Academy of Sciences.

## Antibodies and reagents

Anti-Nono (Cat# 11058-1-AP and 66361-1-Ig) was from Proteintech. Anti-Elk3 (Cat# NBP1-83960) was from Novus Biologicals. Anti-Tmem241 (Cat# 203644-T32) was from Sinobiological. Anti-Lineage cocktail (Cat# 88-7772-72), Anti-CD127 (A7R34), anti-Sca-1 (D7), anti-Flt3 (A2F10), anti-α$_4$β$_7$ (DATK32), anti-Id2 (ILCID2), anti-PLZF (Mags.21F7), anti-Eomes (Dan11mag), anti-NKp46 (29A1.4), anti-NK1.1 (PK136), anti-CD45 (30-F11), anti-CD25 (PC61.5), anti-Gata3 (TWAJ), anti-RORγt (AFKJS-9), anti-Bcl11b (1F8H9), anti-CD3 (17A2), anti-CD19 (1D3), anti-KLRG1 (2F1), anti-CD90 (HIS51), anti-IL-22 (IL22JOP), anti-Ki67 (SolA15), anti-CD45.2 (104), anti-BrdU (BU20A), anti-PD-1 (J43) and anti-CD45.1 (A20) were purchased from eBiosciences (San Diego, USA). Anti-c-Kit (2B8), anti-CD150 (TC15-12F12.2), anti-CD48 (HM48-1), and anti-CD49a (HMα1) were purchased from Biolegend (California, USA). Anti-β-actin (Cat# RM2001) was purchased from Beijing Ray Antibody Biotech. Paraformaldehyde (PFA) and 4',6-diamidino-2-phenylindole (DAPI) were from Sigma. IL-22 ELISA kit was purchased from Neobioscience.

## Generation of knockout mice and cells by CRISPR/Cas9 technology

For a generation of *circTmem241*$^{-/-}$ and *Elk3*$^{-/-}$ mice, CRISPR-mediated single-stranded oligodecxynucleotides donors were synthesized as previously described[52]. Zygotes from C57BL/6 mice were injected with sgRNAs and subsequently transferred to the uterus of pseudo-pregnant females from which viable founder mice were obtained. For a generation of *Nono*$^{-/-}$ and *Ash1l*$^{-/-}$ cells, sgRNAs targeting *Nono* or *Ash1l* were synthesized and cloned into AAV-delivering vectors and infected CHILPs or ILCPs isolated from *Vav-Cre;Cas9-KI* mice. Genomic DNA mutation was identified by PCR screening and DNA sequencing, followed by Western blotting or Northern blotting. sgRNA sequences were listed in Supplementary Table 2. *Vav-Cre*, *Cas9-KI,* and *PLZF*$^{GFPcre}$ mice were purchased from the Jackson Laboratory. All the mouse strains were C57BL/6 background and maintained under specific pathogen-free conditions with approval by the Institutional Committee of the Institute of Biophysics, Chinese Academy of Sciences. This study is compliant with all relevant ethical regulations regarding animal research.

## In vitro differentiation assay

ILC in vitro differentiation assay was performed as previously described[53]. Briefly, cells from femurs were flushed out using phosphate-buffered saline (PBS) containing 5% FBS and filtered through 70 μm strainers. Collected cells were treated with RBC lysis buffer (Tiangen, Beijing) to exclude red cells. CHILPs (Lin$^-$CD25$^-$CD127$^+$Flt3$^-$α4β7$^+$) or ILCPs (Lin$^-$CD25$^-$CD127$^+$ Flt3$^-$α4β7$^+$c-Kit$^+$ PD-1$^+$) were sorted by FACSAria III instrument. OP9-DL1 cells were maintained in complete αMEM medium (supplemented with 10% FBS, 1% streptomycin, 1% penicillin). Before seeding of progenitor cells, OP9-DL1 cells were treated with 4 μg/ml mitomycin for 2 h to inhibit cell division. Then, cells were digested and plated at the density of $1 \times 10^6$ cells per 24-well plate. After OP9-DL1 cells were adhered, indicated progenitor cells were inoculated on OP9-DL1 feeder cells in a complete RIPM1640 medium (Supplemented with 10% FBS, 1% streptomycin, 1% penicillin, 20 ng/ml IL-7, 20 ng/ml SCF). Progenitor cells were detected after culture for 7 days while differentiated ILCs were analyzed 14 days later by FACS.

## Intestinal lymphocyte separation

Intestinal lymphocytes were separated as described before[27]. Briefly, intestines were cut open longitudinally and washed using PBS several times to clean intestinal contents. Then, intestines were cut into 2–5 mm pieces, washed with a solution I buffer (10 mM HEPES and 5 mM EDTA in HBSS) five times to remove epithelial cells, followed by lamina propria digestion with solution II buffer (DNase I, 5% FBS, 0.5 mg/ml collagenase II and collagenase III) three times. Finally, the digested lamina propria lymphocytes (LPLs) were filtered through 70 μm strainers and used for downstream experiments.

## Flow cytometry

For flow cytometric analysis, HSC (Lin$^-$Sca-1$^+$c-Kit$^+$CD150$^+$CD48$^-$), MPP (Lin$^-$Sca-1$^+$c-Kit$^+$CD150$^-$CD48$^+$), CLP (Lin$^-$CD127$^+$c-Kit$^{int}$Sca-1$^{int}$Flt3$^+$α$_4$β$_7^-$), α$_4$β$_7^+$ CLP (Lin$^-$CD127$^+$c-Kit$^{int}$Sca-1$^{int}$Flt3$^+$α$_4$β$_7^+$), CHILP (Lin$^-$CD25$^-$CD127$^+$Flt3$^-$α4β7$^+$Id2$^+$), ILCP (Lin$^-$CD127$^+$Flt3$^-$c-Kit$^+$α4β7$^+$PLZF$^{GFP}$), ILC1P (Lin$^-$CD127$^+$Eomes$^-$CD49a$^+$NK1.1$^+$NKp46$^+$), ILC2P (Lin$^-$CD127$^+$CD45$^+$Flt3$^-$CD117$^-$Sca-1$^+$CD25$^+$Gata3$^+$), ILC3P (Lin$^-$CD25$^-$CD127$^+$α4β7$^{int}$Gata3$^{lo}$Bcl11b$^+$Id2$^+$RORγt$^+$), siILC1 (CD3$^-$CD19$^-$CD127$^+$CD45$^+$NK1.1$^+$NKp46$^+$), siILC2 (CD3$^-$CD19$^-$CD127$^+$CD90$^+$KLRG1$^+$Sca-1$^+$), siILC3 (Lin$^-$CD127$^+$CD45$^+$RORγt$^+$), NK1.1$^+$ NK, CD19$^+$ B and CD3$^+$ T populations were analyzed or sorted with a FACSAria III instrument (BD Biosciences). *PLZF*$^{GFPcre}$ mice were used for ILCP (Lin$^-$CD127$^+$α$_4$β$_7^+$PLZF$^{GFP}$) isolation and *Id2*$^{+/GFP}$ mice were used for CHILP (Lin$^-$CD25$^-$CD127$^+$Flt3$^-$α4β7$^+$Id2$^{GFP}$) isolation by FACS. Data were collected with FACSAria IIIu instrument and analyzed by FlowJo Version 10.0 software.

## IF staining

Intestine tissues were collected and fixed using 4% PFA (Sigma-Aldrich) for one day at room temperature followed by frozen sectioning. Intestine sections were further fixed in 4% PFA for 1 hour at room temperature and permeabilized with PBS containing 1% TritonX-100 for 1 hour, blocked with 10% donkey serum for 1 h at

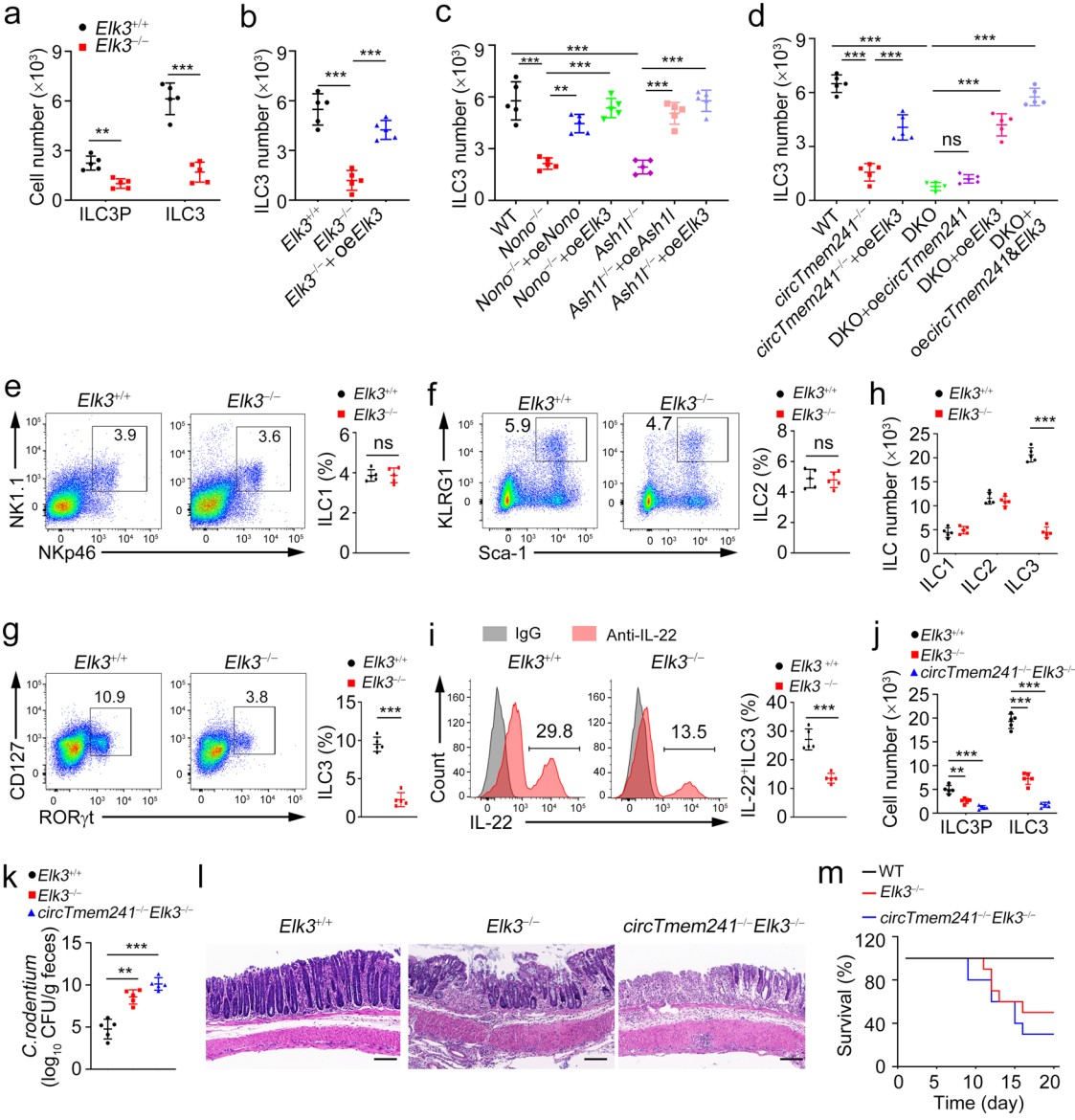

**Fig. 7 | *Elk3* knockout impairs ILC3 commitment and anti-bacterial immunity.**
**a**–**d** Indicated ILCPs were isolated and conducted in vitro differentiation assay with OP9-DL1 feeder cells. ILC3Ps were analyzed 7 days later and ILC3s were analyzed 14 days later by FACS. *n* = 5 for each group. **e**–**g** ILC1s (CD3⁻CD19⁻CD45⁺CD127⁺ gated), ILC2s (CD3⁻CD19⁻CD127⁺CD90⁺ gated) and ILC3s (CD3⁻CD19⁻CD45ˡᵒ gated) were analyzed in small intestines of *Elk3*⁺/⁺ and *Elk3*⁻/⁻ mice by FACS. *n* = 5 for each group. **h** Numbers of indicated ILCs in **e**–**g** were calculated. *n* = 5 for each group. **i** LPLs from *Elk3*⁺/⁺ and *Elk3*⁻/⁻ mouse intestines were stimulated by IL-23 for 4 h, followed by IL-22⁺ ILC3 detection with FACS. **j** ILC3Ps and ILC3s from *Elk3*⁺/⁺, *Elk3*⁻/⁻,

and *circTmem241*⁻/⁻*Elk3*⁻/⁻ mice were analyzed by FACS, and indicated cell numbers were calculated. *n* = 5 for each group. **k** CFUs in feces of *Elk3*⁺/⁺, *Elk3*⁻/⁻ and *circTmem241*⁻/⁻*Elk3*⁻/⁻ mice were measured after *C. rodentium* infection for 6 days. *n* = 5 for each group. **l** Colon tissues from *Elk3*⁺/⁺, *Elk3*⁻/⁻ and *circTmem241*⁻/⁻*Elk3*⁻/⁻ mice were analyzed by H&E staining. Scale bars, 100 μm. **m** Survival rates of indicated mice were measured after infection with *C. rodentium*. *n* = 10 for each group. **P* < 0.01 and ***P* < 0.001. Data were analyzed by an unpaired two-side Student's *t* test and shown as means ± SD. Data are representative of at least three independent experiments. Source data are provided as a Source Data file.

---

room temperature, incubated with appropriate primary antibodies at 4 °C overnight, and then incubated with fluorescence-conjugated secondary antibodies. For ILC3 staining, anti- RORγt (host species: rat) was used at a dilution of 1:100, and Alexa-Flour 594 donkey anti-rat secondary antibody at 1:500. Anti-CD127/IL-7R (host species: rabbit) was used at 1:200 and Alexa-Flour 647 donkey anti-rabbit secondary antibody at 1:500. FTIC-conjugated anti-CD3 was used at 1:300 directly. For ILCP staining, anti-Nono (host species: mouse) was used at a dilution of 1:200 and Alexa-Flour 594 donkey anti-mouse secondary antibody at 1:500. DAPI was used for nucleus staining. Sections were visualized with Nikon A1R+ confocal microscopy.

**Fluorescence in situ hybridization**
CHILPs (Lin⁻CD25⁻CD127⁺Flt3⁻α4β7⁺), ILCPs (Lin⁻CD25⁻CD127⁺Flt3⁻α4β7⁺c-Kit⁺PD-1⁺) or ILC3s (Lin⁻CD127⁺CD90ʰⁱᵍʰCD45ˡᵒ) were isolated and spread on high-adherent slides. Cells were fixed using 4% PFA (Sigma-Aldrich) for 20 min at room temperature, permeabilized with PBS containing 1% TritonX-100 for 20 min, prehybridized with hybridization buffer (50% formamide, 5× SSC, 500 ng/μl yeast tRNA, 1× Dehardt's solution, 500 ng/μl sperm DNA, 50 ng/μl Heparin, 2.5 mM EDTA, 0.1% Tween-20, 0.25% CHAPS) for 1 h at 45 °C, incubated with biotinylated or cy5-labeled probes at 45 °C for 2 hours, and then washed three times with SSC washing buffer. After blocking with 10% donkey serum, cells were subjected to IF staining and visualized with Nikon A1R + confocal microscopy.

## DNA in situ hybridization

DNA in situ hybridization was conducted according to a previous study with minor modification[54]. In Brief, ILCPs were spread on adherent slides and fixed by 4% paraformaldehyde (PFA, Sigma-Aldrich), and quenched in 0.1 M Tris-HCl (pH 7.4). After pre-metallized with 0.1% TritonX-100, slides were incubated in PBS containing 20% glycerol for 20 min. Then slides were subjected to 3 freeze/thawing cycles in liquid nitrogen. after which cells were equilibrated in hybridization buffer (50% formamide, 2× SSC) for 10 min. Biotin-labeled probe against *Elk3* promoter was pipetted onto slides and incubated at 78 °C for exact 2 min, followed by incubation at 37 °C overnight. Cells were subjected to IF staining with indicated antibodies the next day. DAPI was used for nucleus staining. Cells were visualized with Nikon A1R + confocal microscopy.

## Real-time quantitative PCR

Total RNA was isolated through a RNA Miniprep Kit (Tiangen, Beijing, China) following the manufacturer's instructions. cDNA was synthesized using 5× All-In-one RT Mastermix (Abm, Vancouver, Canada) and analyzed on QuantStudio1 qPCR system using specific primer pairs listed in Supplementary Table 3. The relative expression level was calculated and normalized to endogenous *18 S* or *Gapdh*. CircRNAs were analyzed using specific divergent and convergent primers (Supplementary Table 4).

## CHIRP assay

CHIRP assay was performed according to previous research[55]. ILCPs were cross-linked with 1% glutaraldehyde at room temperature for 10 min and quenched by 0.1 M glycine for 5 min. Then cells were washed twice with PBS, lysed with Lysis buffer (1% SDS, 10 mM EDTA, 50 mM Tris, supplemented with PMSF, proteinase inhibitor cocktails, and RNase inhibitor) and sonicated to produce 200-500 bp DNA fragments. Lysates were incubated with biotinylated antisense probes against *circTmem241* junction site for 4 hours at 37 °C. Streptavidin agarose beads were then added to isolate probe binding complex. After washing steps, DNA, RNA, or proteins were eluted from beads and subjected to downstream analysis.

## ChIP assay

ILCPs were cross-linked with 1% formaldehyde at 37 °C for 10 min. Then cells were washed twice with PBS, lysed with SDS lysis buffer (1% SDS, 10 mM EDTA, 50 mM Tris), and sonicated to make 200 to 500 bp DNA fragments. Lysates were pre-cleared with Protein A Agarose/Salmon Sperm DNA (50% Slurry) and then incubated with 4 μg indicated antibodies overnight at 4 °C. After washing steps, DNA was eluted from beads and purified. DNA fragments were analyzed using primer pairs listed in Supplementary Table 5.

## Luciferase reporter assay

For luciferase reporter assay, the full-length promoter of *Elk3* (from −2000 to 0 upstream from transcriptional start site) or truncated *Elk3* promoter was cloned into pGL3 vector. For mutation of *Elk3* promoter, the pairing region was substituted with mutated sequence (5′-AGAAGTTCCATAAT-3′) in full-length promoter. Exon region of *circTmem241* with flanking complementary elements was constructed into pcDNA4 vector for overexpression. 293 T cells were seeded in 24-well plate one day before transfection. For transfection of each well, 100 ng pGL3 plasmid, 1 ng pRL-TK and 500 ng pcDNA4 plasmid were used. Cells were transferred to 96-well assay plate at 10000/well 24 h later. After cells were attached, luminescence signals were analyzed using ONE-Glo Luciferase Assay System (Promega, Madison) according to the manufacturer's protocol.

## C. rodentium infection

Mice were fasted for 8 h and then infected with $5 \times 10^9$ *C. rodentium* by intragastric gavage. *C. rodentium* was a gift from B. Ge (Shanghai Institutes for Biological Sciences, Chinese Academy of Sciences). After infection for seven days, mice were sacrificed and colons were collected for pathological analysis. Feces were analyzed for bacterial loads. LPLs were isolated for analyses of ILC3s and IL-22 production by FACS and enzyme-linked immunosorbent assay (ELISA).

## Histopathological analysis

Colons from indicated mice were cut open longitudinally and fixed in 4% PFA followed by paraffin sectioning and H&E staining. The histopathological score was evaluated and analyzed according to a previous study[56]. Briefly, sections were assessed for epithelial hyperplasia (calculated based on percentage change above control, 0 for no change, 1 for 1–50%, 2 for 51-100%, 3 for more than 100%), epithelial integrity (0 for no change, 1 for <10 epithelia shedding per lesion, 2 for 11–20 shedding per lesion, 3 for epithelia ulceration, 4 for sever crypt destruction), granulocyte mononuclear cell infiltration (0 for none, 1 for mild, 2 for moderate, 3 for severe), depletion of goblet cells (goblet cells from ×400 magnification filed were calculated, 0 for >50, 1 for 25–50, 2 for 10–25, 3 for <10) and submucosal oedema (0 for no change, 1 for mild, 2 for moderate, 3 for severe).

## Bone marrow transplantation

For non-competitive transplantation, $5 \times 10^6$ CD45.2$^+$ BM donor cells from indicated mice were injected into lethally irradiated CD45.1$^+$ recipient mice. Eight weeks after transplantation, CD45.2$^+$ ILC3s derived from donor cells were analyzed by cytometric analysis. For competitive transplantation, $1 \times 10^6$ CD45.2$^+$ BM cells from indicated mice and $1 \times 10^6$ wild-type CD45.1$^+$ BM cells were mixed and transplanted into lethally irradiated CD45.1$^+$ recipient mice. Eight weeks after transplantation, the ratio of CD45.2$^+$ ILC3s to CD45.1$^+$ ILC3s were examined.

## Plasmids construction and virus preparation

RNA interference was performed as previously described[27]. And target sequences were listed in Supplementary Table 6. For *circTmem241* overexpression, genomic exon region of *circTmem241* was constructed into pcDNA4 vector flanked by the upstream and downstream complementary elements. Lentiviral vector (pSIN-EF2-GFP) was co-transfected with packaging plasmids pMD2G and psPAX2 into HEK293T cells for 48 h, followed by culture medium collection and ultracentrifugation at 25,000×*g* for 2 h. Pellets were resuspended in DMEM medium and viral titers were determined by infecting HEK293T cells with diluted viruses. Cells were incubated with lentiviruses and centrifuged at 500×*g* for 2 h in the presence of 8 μg/ml polybrene. Cells were cultured for 24 h to allow GFP expression, followed by sorting of GFP positive cells through a flow cytometer. HEK293T cells were from ATCC (CRL-11268) and tested negative for mycoplasma contamination.

## RNA-pulldown assay

Bone marrow cells were lysed and supernatants were incubated with biotin-labeled *circTmem241* or antisense control probe. Then precipitated components were separated with SDS-PAGE and silver staining. Differential bands enriched by *circTmem241* were analyzed by mass spectrometry or western blot.

## Transcriptome analysis

For transcriptome RNA-seq analysis, ILCPs were isolated from *circTmem241*$^{+/+}$ or *circTmem241*$^{-/-}$ mice. Then RNAs were isolated using Trizol reagent (Invitrogen) and subjected to RNA sequencing by Beijing Genomics Institute. Subsequent data analysis was conducted by R studio and GSEA software.

## Electrophoretic mobility shift assay

For analysis of the interaction between *circTmem241* and Nono, biotin-labeled linearized *circTmem241* was synthesized by in vitro transcription and Nono was expressed and purified by GST-tag. *CircTmem241* and Nono were incubated in EMSA reaction buffer and mobility shift assay was performed using a Light Shift Chemiluminescent RNA EMSA Kit (Thermo Scientific) according to the manufacturer's protocol.

## Nuclear run-on assay

Nuclear run-on assay was performed as previously described[27]. ILCPs isolated from indicated mice were suspended in nuclear extraction buffer (10 mM Tris-HCl, 150 mMKCl, 4 mM MgOAc, pH 7.4), followed by centrifugation to collect cell pellets. Then, pellets were lysed by lysis buffer (nuclear extraction buffer supplemented with 0.5% NP-40), followed by sucrose density gradient centrifugation to prepare transcriptional active crude nuclei components. Crude nuclei were incubated with a biotin labeling mix (Roche, Basel, Swiss) and RNase inhibitor at 28 °C for 5 min. RNAs were extracted using TRIzol reagent according to the manufacturer's instructions. DNA was digested by DNase I for 15 min at room temperature. RNA transcripts were enriched with streptavidin affinity beads, followed by reverse transcription and RT-PCR analysis.

## Enzyme-linked immunosorbent assay

LPLs or ILC3s were isolated and cultured for 24 h with indicated cytokines. Then supernatants were collected and cytokines were detected using ELISA kit (Neobioscience) according to the manufacturer's instructions.

## Statistical and Reproducibility

For statistical evaluation, an unpaired Student's *t* test was applied for calculating statistical probabilities in this study. For all panels, at least three independent experiments were performed with similar results, and representative experiments are shown. Data were analyzed by using Microsoft Excel 2016 or GraphPad Prism Version 8.3. *P* values ≤0.05 were regarded as statistical significance.

## Reporting summary

Further information on research design is available in the Nature Research Reporting Summary linked to this article.

## Data availability

The source data for Figs. 1–7 and Supplementary Figs. 1–6 are provided as a Source Data file. The RNA sequencing data generated in this study have been deposited in the GEO database under accession code GSE201065. All other data are included in the supplemental information or available from the authors upon reasonable requests. Source data are provided with this paper.

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

## Acknowledgements

We thank Jiajia Hou, Zixin Zhao, Yihui Xu, and Junying Jia for technical support. We thank Jing Li (Cnkingbio Company Ltd, Beijing, China) for his technical support. We also thank professor Liu (Academy of Military Medical Sciences) for providing the OP9-DL1 cell line. This work was supported by the National Key R&D Program of China (2019YFA0508501, 2020YFA0803501, 2021YFA1302000), National Natural Science Foundation of China (31930036, 82130088, 81921003, 92042302, 32070533, 91940305, 81772646, 32170874, 31870883), Strategic Priority Research Programs of the Chinese Academy of Sciences (XDB19030203).

## Author contributions

N.L., J.H., and D.F. performed experiments; N.L. designed the project, analyzed the data, and wrote the paper; D.F. and X.Z. constructed genetic mouse strains. N.L., J.H., Y.G., J.W., H.L., and Y.D. analyzed data; Y.T. generated animal models and analyzed data; Z.F. and B. L. initiated the study, organized, designed, and wrote the paper.

## Competing interests

The authors declare no competing interests.
