## [Peer Review File · Nature Communications]

Circular RNA circTmem241 drives group III innate lymphoid cell differentiation via initiation of Elk3 transcriptionREVIEWER COMMENTS

Reviewer #1 (Remarks to the Author):

Liu et al. demonstrated that circTmem241-Nono-Ash11-ELK3 axis in ILCP drives ILC3 differentiation. They first demonstrated that circTmem241 knockout mouse have less ILC3 and susceptible to bacterial infection. They confirmed that circTmem241 directly regulates ILC3P commitment and ILC3 differentiation. By performing transcriptome analysis of ILCP isolated from circTmem241 knockout mouse they identified ELK3 as a downstream target of circTmem241. They conducted molecular analysis to demonstrate the binding of circTmem241 on the ELK3 promoter to promote ELK3 transcription. By performing RNA pulldown assay and immunoprecipitation assay, they identified Nono protein and Ash11 (histone methyltransferase), as an epigenetic regulator of ELK3 promoter and adaptor protein of circTmem241, respectively. Finally, they demonstrated that ELK3 knockout mouse have less ILC3 and susceptible to bacterial infection.

Though their flow is scientifically logical and most of their data, especially molecular data, are solid enough to support their conclusion, there is one major disconnection in their conclusion. Additionally, there are other points, statistical analysis of major data, for example, which should be improved.

Detailed comments;

<Major>

(1) Their conclusion is that circTmem241-Nono-Ash11-ELK3 axis is required for the ILCP differentiation into ILC3P and ILC3 maturation (page 2). Since the end product of the axis is ELK3, the result of ELK3 rescue in circTmem241 knockout mouse needs to be presented in figure-7.

(2) Page 5. Supplementary Fig.2D showed that CHILPs and ILCPs also have circTmem241 expression in the nucleus. (a) Please clarify the point at the manuscript (b) Please display RNA abundance data of circTmem241 in CHILP and ILCP at supplementary Fig.2E.

(3) Figure 1H. It is not clear which spot the yellow arrow indicates. It would be very helpful to understand the image if the authors magnify major part of the image.

(4) Page 7, Figure 2I-J. (a) The manuscript does not describe the main point of this data, which is the rescue of ILC3P and ILC3 differentiation by ectopic expression of circTmem241 in circTmem241^{-/-} in vitro. (b) The statistical significance of the data should be revised to stress the point.

(5) Page 8, Figure 4G and Supple Fig-5B. Material and method for figure 4G is missing. For example, which part is the full length of ELK3 promoter? Which cell line did the author use for the reporter assay? More importantly, luciferase assay with Δ -200~0 promoter does not support the binding of circTmem241 on the region of ELK3 promoter. Since the authors identified the binding site of circTmem241 on the ELK3 promoter, they can demonstrate the circTmem241 mediated regulation of ELK3 promoter activity by performing luciferase assay with ELK3 mutant promoter.

(6) Figure 4L: The statistical significance of the data should be revised to stress the rescue of ILC3 commitment by the overexpression of WT or mutant form of circTmem241.

(7) Figure 5F: It is not clear which spot is colocalization point of circTmem241 with Nono. It would be very helpful to understand the image if the authors magnify major part of the image .

(8) Figure 7B, 7C, 7D: The statistical significance of the data should be revised to stress the rescue of ILC3 commitment by the overexpression of ELK3.

<Minor>

(1) p5, Figure 1A. Since shcircAmotl1 also significantly impaired ILC3 differentiation, 'only

- circTmem241 knockdown significantly impaired ILC3 differentiation' is not appropriate.
- (2) Detailed description is required for the figure legend of Figure 3D-F. Are these results came from competitive transplantation??
- (3) Supplementary Figure 4G: Is that ELK3 overexpression performed in circTmem241^{-/-} ILC3? Compared to Fig 4D, ELK3 expression in vector lane seems to be too high.
- (4) Supplementary Figure 4H: The statistical significance of the data should be revised to stress the rescue of ILC3 commitment by ELK3 overexpression.

Reviewer #2 (Remarks to the Author):

Liu et al report the identification of a new circular RNA (circTmem241) that is required for ILC3 lineage commitment. The authors also uncover a new role for the transcription factor Elk3 in ILC3 differentiation. The authors generated strains of knockout mice to probe the function of circTmem241 and Elk3 in ILC3s in vivo. Furthermore, using multiple molecular biology and biochemical approaches, the authors define interaction partners and the molecular mechanism of how circTmem241 regulates ILC3 lineage commitment. This is a study based on well-designed and carefully executed experiments, some of which are technically difficult. The conclusions are supported by the experimental data and the manuscript is well-written. The study provides important new insights into the molecular drivers of ILC3 differentiation from ILC precursors. Regulation of immune function by circular RNAs is an emerging area and the findings reported in the manuscript are of interest to the readership of Nature Communications. To further strengthen the study, the following few comments should be addressed.

Major comments

- (1) The authors show that ROR γ ⁺ ILC3s are globally reduced in circTmem241-deficient mice. ILC3 consist of several subsets that can be distinguished by cell surface expression of CCR6 and NKp46. Therefore, does circTmem241 deficiency reduce the number of LT α -like ILC3s (CCR6⁺NKp46⁻), NKR⁺ ILC3s (CCR6⁻NKp46⁺), and/or double negative (CCR6⁻NKp46⁻) ILC3s in the small intestine?
- (2) The authors demonstrate that circTmem241 promotes IL-22 production by ILC3s and host defense against infection with intestinal bacteria. Another major function of ILC3s is the formation of lymphoid tissues in the intestine. Therefore, is the development of cryptopatches, isolated lymphoid follicles, Peyer's patches impaired in mice lacking circTmem241 or Elk3?
- (3) How is circTmem241 regulated in ILC3s? Is circTmem241 upregulated by signals that induce ILC3 differentiation? Considering its high expression in the intestine, do gut-specific signals such as nutrients and metabolites control circTmem241 formation? Tmem241 is a predicted sugar transport protein, raising the idea that nutrient availability may regulate the formation of circTmem241. It is known that nutrient status affects the ILC3/ILC2 ratio in the intestine (Spencer et al, Science 2014). These possibilities should at least be discussed in the manuscript.

Minor comments

- (1) Does circTmem24 promote the production of IL-17 by ILC3s?

Point-by-point response to the reviewers

Reviewer #1:

Liu et al. demonstrated that circTmem241-Nono-Ash1-ELK3 axis in ILCP drives ILC3 differentiation. They first demonstrated that circTmem241 knockout mouse have less ILC3 and susceptible to bacterial infection. They confirmed that circTmem241 directly regulates ILC3P commitment and ILC3 differentiation. By performing transcriptome analysis of ILCP isolated from circTmem241 knockout mouse they identified ELK3 as a downstream target of circTmem241. They conducted molecular analysis to demonstrate the binding of circTmem241 on the ELK3 promoter to promote ELK3 transcription. By performing RNA pulldown assay and immunoprecipitation assay, they identified Nono protein and Ash1 (histone methyltransferase), as an epigenetic regulator of ELK3 promoter and adaptor protein of circTmem241, respectively. Finally, they demonstrated that ELK3 knockout mouse have less ILC3 and susceptible to bacterial infection.

Though their flow is scientifically logical and most of their data, especially molecular data, are solid enough to support their conclusion, there is one major disconnection in their conclusion. Additionally, there are other points, statistical analysis of major data, for example, which should be improved.

Major comments:

(1) Their conclusion is that circTmem241-Nono-Ash1-ELK3 axis is required for the ILCP differentiation into ILC3P and ILC3 maturation (page 2). Since the end product of the axis is ELK3, the result of ELK3 rescue in circTmem241 knockout mouse needs to be presented in figure-7.

Answer: This is a good suggestion. We performed the *in vitro* rescue differentiation assay in *circTmem241*^{-/-} ILCPs. We found that Elk3 overexpression in *circTmem241*^{-/-} ILCPs was able to rescue ILC3 commitment (new Fig. 7d).

(2) Page 5. Supplementary Fig.2D showed that CHILPs and ILCPs also have circTmem241 expression in the nucleus. (a) Please clarify the point at the manuscript (b) Please display RNA abundance data of circTmem241 in CHILP and ILCP at supplementary Fig.2E.

Answer: We measured the *circTmem241* abundance of nuclear and cytoplasmic fractions in both CHILPs and ILCPs (new Supplementary Fig. 2e). We also revised our manuscript to clarify that circTmem241 is mainly distributed in the nuclei of CHILPs and ILCPs.

(3) Figure 1H. It is not clear which spot the yellow arrow indicates. It would be very helpful to understand the image if the authors magnify major part of the image.

Answer: We magnified the major part of Fig. 1h to display ILC3s that the yellow arrows indicate (new Fig. 1h).

(4) Page 7, Figure 2I-J. (a) The manuscript does not describe the main point of this data, which is the rescue of ILC3P and ILC3 differentiation by ectopic expression of circTmem241 in circTmem241^{-/-} in vitro. (b) The statistical significance of the data should be revised to stress the point.

Answer: We performed the statistical analysis in the new Fig. 2i-j and stated the rescue of ILC3P and ILC3 differentiation by ectopic expression of *circTmem241* in *circTmem241*^{-/-} in vitro in our revised manuscript.

(5) Page 8, Figure 4G and Supple Fig-5B. Material and method for figure 4G is missing. For example, which part is the full length of ELK3 promoter? Which cell line did the author use for the reporter assay? More importantly, luciferase assay with Δ -200~0 promoter does not support the binding of *circTmem241* on the region of ELK3 promoter. Since the authors identified the binding site of *circTmem241* on the ELK3 promoter, they can demonstrate the *circTmem241* mediated regulation of ELK3 promoter activity by performing luciferase assay with ELK3 mutant promoter.

Answer: We described the material and method of luciferase reporter assay in detail in the Methods section. We mutated the binding site in *Elk3* promoter and performed luciferase reporter assay. We found that mutation in *Elk3* promoter abrogated the luciferase activity after *circTmem241* overexpression (new Supplementary Fig. 5c).

(6) Figure 4L: The statistical significance of the data should be revised to stress the rescue of ILC3 commitment by the overexpression of WT or mutant form of *circTmem241*.

Answer: We performed the statistics in the new Figure 4l.

(7) Figure 5F: It is not clear which spot is colocalization point of *circTmem241* with Nono. It would be very helpful to understand the image if the authors magnify major part of the image.

Answer: We enlarged this figure to display colocalization of *circTmem241* with Nono (new Fig. 5f).

(8) Figure 7B, 7C, 7D: The statistical significance of the data should be revised to stress the rescue of ILC3 commitment by the overexpression of ELK3.

Answer: We performed the statistical significance in Figure 7b-d accordingly.

<Minor>

(1) p5, Figure 1A. Since *shcircAmotl1* also significantly impaired ILC3 differentiation, 'only *circTmem241* knockdown significantly impaired ILC3 differentiation' is not appropriate.

Answer: We corrected this statement in our revised manuscript.

(2) Detailed description is required for the figure legend of Figure 3D-F. Are these results came from competitive transplantation??

Answer: We described the figure legends of Figure 3d-f in detail. These results were derived from non-competitive transplantation reconstructed mice.

(3) Supplementary Figure 4G: Is that ELK3 overexpression performed in *circTmem241*^{-/-} ILCP? Compared to Fig 4D, ELK3 expression in vector lane seems to be too high.

Answer: This is the case. We overexpressed *Elk3* in *circTmem241*^{-/-} ILCPs for this assay. The image of supplementary Fig. 4g could expose a longer time. We repeated this

assay and changed a better image.

(4) Supplementary Figure 4H: The statistical significance of the data should be revised to stress the rescue of ILC3 commitment by ELK3 overexpression.

Answer: We performed the statistical significance in Supplementary Figure 4h.

Reviewer #2:

Liu et al report the identification of a new circular RNA (*circTmem241*) that is required for ILC3 lineage commitment. The authors also uncover a new role for the transcription factor *Elk3* in ILC3 differentiation. The authors generated strains of knockout mice to probe the function of *circTmem241* and *Elk3* in ILC3s *in vivo*. Furthermore, using multiple molecular biology and biochemical approaches, the authors define interaction partners and the molecular mechanism of how *circTmem241* regulates ILC3 lineage commitment. This is a study based on well-designed and carefully executed experiments, some of which are technically difficult. The conclusions are supported by the experimental data and the manuscript is well-written. The study provides important new insights into the molecular drivers of ILC3 differentiation from ILC precursors. Regulation of immune function by circular RNAs is an emerging area and the findings reported in the manuscript are of interest to the readership of *Nature Communications*. To further strengthen the study, the following few comments should be addressed.

Major comments

(1) The authors show that ROR γ t⁺ ILC3s are globally reduced in *circTmem241*-deficient mice. ILC3 consist of several subsets that can be distinguished by cell surface expression of CCR6 and NKp46. Therefore, does *circTmem241* deficiency reduce the number of LTi-like ILC3s (CCR6+NKp46⁻), NKR⁺ ILC3s (CCR6-NKp46⁺), and/or double negative (CCR6-NKp46⁻) ILC3s in the small intestine?

Answer: This is a good point. We analyzed different ILC3 subsets in *circTmem241*-deficient mice and littermate control using FACS. We found that all three subsets were decreased in *circTmem241*-deficient mice (new Supplementary Fig. 3f).

(2) The authors demonstrate that *circTmem241* promotes IL-22 production by ILC3s and host defense against infection with intestinal bacteria. Another major function of ILC3s is the formation of lymphoid tissues in the intestine. Therefore, is the development of cryptopatches, isolated lymphoid follicles, Peyer's patches impaired in mice lacking *circTmem241* or *Elk3*?

Answer: This is a very good point. We analyzed the numbers of Peyer's patches, cryptopatches and isolated lymphoid follicles in mice lacking *circTmem241* or *Elk3* compared with their littermate control mice. We found that the development of these gut-associated lymphoid tissues were impaired after *circTmem241* or *Elk3* knockout (new Supplementary Fig. 3g and 6h).

(3) How is *circTmem241* regulated in ILC3s? Is *circTmem241* upregulated by signals that induce ILC3 differentiation? Considering its high expression in the intestine, do gut-specific signals such as nutrients and metabolites control *circTmem241* formation?

Tmem241 is a predicted sugar transport protein, raising the idea that nutrient availability may regulate the formation of circTmem241. It is known that nutrient status affects the ILC3/ILC2 ratio in the intestine (Spencer et al, Science 2014). These possibilities should at least be discussed in the manuscript.

Answer: We isolated ILC3s from intestinal LPLs and stimulated with different nutrient components including retinoic acid, propionate, glucose and fructose, followed by qRT-PCR analysis of *circTmem241* expression. We found that nutrient availability did not affect circTmem241 formation (Attached Fig. 1). How circTmem241 was regulated in ILC3s needs to be further investigated. We discussed this issue in the Discussion section.

Minor comments

(1) Does circTmem24 promote the production of IL-17 by ILC3s?

Answer: We stimulated LPLs isolated from *circTmem241*-deficient mice and wild-type mice for 4 hours and analyzed IL-17 production by FACS analysis. We found that IL-17A production was impaired in *circTmem241*-deficient ILC3s (Attached Fig. 2).

Attached Figure 1. Nutrient availability did not affect circTmem241 formation. Relative expression of *circTmem241* was measured in ILC3s after stimulation with different nutrient components with qPCR. Fold changes were normalized to endogenous 18S.

Attached Figure 2. IL-17A production is decreased in *circTmem241*-deficient ILC3s. Lamina propria lymphocytes (LPLs) from *circTmem241*^{+/+} and *circTmem241*^{-/-} mouse intestines were sorted and stimulated by IL-23 for 4 h, followed by IL-17⁺ ILC3 detection with FACS.

REVIEWERS' COMMENTS

Reviewer #1 (Remarks to the Author):

All of reviewer-1's concerns were resolved.

Reviewer #2 (Remarks to the Author):

The authors have sufficiently addressed all my comments.

Point-by-point response to the reviewers

Reviewer #1:

All of reviewer-1's concerns were resolved.

Answer: Thanks for your insightful suggestions and positive comments.

Reviewer #2:

The authors have sufficiently addressed all my comments.

Answer: Thanks for your comprehensive questions and positive comments.